_Article_

# JAK/STAT inhibition protects glucocorticoid receptor knockout mice from lethal malaria-induced hypoglycemia and hyperinflammation

Fran Prenen [1,7], Leen Vandermosten [1,7], Sofie Knoops[1], Emilie Pollenus[1], Hendrik Possemiers[1], Pauline Dagneau de Richecour[1], Giorgio Caratti [2,3], Christopher Cawthorne [4], Sabine Vettorazzi[2], Yevva Cranshoff [1], Dominique Schols [5], Sandra Claes[5], Christophe M Deroose [4], Uwe Himmelreich [6], Jan Tuckermann [2] & Philippe E Van den Steen [1]✉

## Abstract

**Disease tolerance is a key defense mechanism that limits damage to the host without directly reducing pathogen levels. In malaria, these mechanisms are essential for preventing severe disease and death but remain poorly understood. In this study, we show that glucocorticoid receptor (GR)-mediated processes play a vital role in disease tolerance during _Plasmodium chabaudi_ AS infection. GR deletion in infected mice resulted in lethal hypoglycemia and a cytokine storm. Hypoglycemia was driven by severe metabolic dysfunction in the liver and spleen, characterized by increased glucose uptake, glycogen depletion, a dominant glycolytic profile and reduced gluconeogenic gene expression. Importantly, this hypoglycemic state was strongly associated with overactivation of the JAK/STAT pathway and excessive cytokine expression. Treatment with the JAK1/2 inhibitor ruxolitinib significantly improved survival by preventing lethal hypoglycemia and suppressing hyperinflammation. Our findings reveal a novel link between GR signaling, STAT3 activation, cytokine expression and glucose metabolism during severe malaria. This underscores the critical role of GR-mediated processes in disease tolerance and highlights ruxolitinib as a promising adjuvant therapy for managing life-threatening metabolic complications in malaria.**

**Keywords** Glucocorticoid Receptor; Hypoglycemia; Malaria; Ruxolitinib; Tolerance
**Subject Category** Microbiology, Virology & Host Pathogen Interaction

## Introduction

Malaria, caused by protozoan parasites of the genus _Plasmodium_, remains one of the deadliest infectious diseases worldwide. In 2023 alone, the World Health Organization reported 263 million clinical malaria cases and 597,000 deaths (World Health Organization, 2023). The clinical outcomes of malaria are highly variable, ranging from mild symptoms with fever and fatigue to severe, life-threatening conditions such as cerebral malaria, severe anemia, acute respiratory distress syndrome (ARDS), acute kidney injury, hypoglycemia, and metabolic acidosis (Deroost et al, 2016). A key concept in understanding these varied outcomes is disease tolerance, a defense strategy which minimizes damage to the host without necessarily reducing the pathogen load (Soares et al, 2017; Medzhitov et al, 2012). This evolutionarily conserved mechanism allows some individuals to tolerate high parasitemia without severe symptoms (Imwong et al, 2016). Research using mouse models already provided significant insights into the critical role of immunometabolic adaptations in fostering this tolerance (Hirako et al, 2018; Wang et al, 2018; Cumnock et al, 2018; Ramos et al, 2022; Vandermosten et al, 2018a).

Glucocorticoids (GCs) have proven to be essential in driving disease tolerance in conditions such as sepsis (Vandewalle and Libert, 2020). GCs are stress hormones produced by the adrenal glands and are tightly regulated by the hypothalamic–pituitary–adrenal (HPA) axis (Timmermans et al, 2019). Upon release, GCs bind to the glucocorticoid receptor (GR), influencing a vast array of cellular processes, from gene transcription to non-genomic actions (De Bosscher and Haegeman, 2009; Vettorazzi et al, 2022; Panettieri et al, 2019). Crucially, GCs regulate immune responses and glucose metabolism, making them indispensable in maintaining homeostasis during infections (Cain and Cidlowski, 2017; Magomedova and Cummins, 2016; Auger et al, 2024; Stifel et al, 2022). Recent studies have underscored the significance of GCs in malaria, showing elevated levels during both human and mouse infections (Vandermosten et al, 2018a; Abdrabou et al, 2021;

[1]Laboratory of Immunoparasitology, Department of Microbiology, Immunology and Transplantation, Rega Institute for Medical Research, KU Leuven, Leuven 3000, Belgium. [2]Institute of Molecular Endocrinology and Physiology, German Center for Child and Adolescent Health (DZKJ), partner site Ulm, Ulm University, Ulm 89081, Germany. [3]Oxford Centre for Diabetes, Endocrinology and Metabolism, NIHR Oxford Biomedical Research Centre, University of Oxford, Churchill Hospital, Oxford OX3 7LE, UK. [4]Nuclear Medicine and Molecular Imaging, Department of Imaging and Pathology, KU Leuven, Leuven 3000, Belgium. [5]Laboratory of Molecular, Structural and Translational Virology, Department of Microbiology, Immunology and Transplantation, Rega Institute for Medical Research, KU Leuven, Leuven 3000, Belgium. [6]Biomedical MRI, Department of Imaging and Pathology, 3000 KU Leuven, Leuven, Belgium. [7]These authors contributed equally: Fran Prenen, Leen Vandermosten. ✉E-mail: philippe.vandensteen@kuleuven.be

Vandermosten et al, 2023). Our own work has demonstrated that adrenal hormones are essential for promoting disease tolerance in experimental malaria (Vandermosten et al, 2018a). We observed reduced survival rates in adrenalectomized mice, which could be rescued by the supplementation of dexamethasone, a synthetic GC. Importantly, lethality in malaria-infected adrenalectomized mice was associated with hypoglycemia, which was independent of insulin and TNF-α. This suggests a completely different mechanism compared to the milder hypoglycemia observed in wild-type (WT) mice with mild malaria or viral infections (Hirako et al, 2018, Šestan et al, 2024). In addition, we recently established a link between diminished GC responses and disease severity in children with malaria (Vandermosten et al, 2023). Yet, the specific mechanisms by which endogenous GCs mediate disease tolerance during malaria remain poorly understood.

In this study, we used tamoxifen-inducible glucocorticoid receptor knockout (GRiKO) mice to dissect the role of GR-mediated processes in experimental malaria. Employing the rodent-specific *Plasmodium chabaudi* AS (*Pc*AS) model, which typically induces only mild symptoms despite high parasitemia in WT mice (Stephens et al, 2012), we found that GR deletion leads to excessive cytokine levels and lethal hypoglycemia, without alteration of insulin levels, highlighting a distinct pathway in glucose home-ostasis. Instead, hypoglycemia was marked by increases in glucagon levels and severe metabolic dysregulation in the liver and spleen. Remarkably, the overactivation of the Janus kinase (JAK)/signal transducer and activator of transcription (STAT) pathway was central to this hypoglycemic phenotype, as its inhibition with ruxolitinib successfully prevented the lethal outcome. These findings reveal a previously unrecognized connection between GR signaling, cytokine expression, JAK/STAT activation and glucose metabolism in malaria. Overall, our results underscore the essential role of GR-mediated processes in disease tolerance and the potential of JAK/STAT inhibition as a therapeutic strategy for managing severe malaria-induced hypoglycemia.

# Results

## Confirmation of GR deletion upon tamoxifen administration

To study the role of endogenous GCs in vivo, we used tamoxifen-inducible GRiKO mice which carry two genetic modifications: a homozygous floxed exon 3 of the GR gene, and a heterozygous tamoxifen-inducible Cre-ERT2 recombinase under the control of the ubiquitously active Rosa26 promotor (Rapp et al, 2018; Tronche et al, 1999). Mice that are Cre-ERT2-positive are hereafter referred to as GRiKO, while Cre-negative littermates are referred as WT, because tamoxifen treatment results in Cre-ERT2 translocation to the nucleus and subsequent excision of the floxed GR exon 3. First, we validated the efficiency of global GR deletion in GRiKO mice upon tamoxifen administration. GRiKO mice (Cre-ERT2[+]) were treated for 5 consecutive days with 10 mg/kg tamoxifen or with vehicle. Spleen, liver, brain and lung tissues were collected 1, 2, or 4 weeks after the final tamoxifen treatment. Western blot analysis showed an 88 to 99% reduction in GRα protein expression in both spleen and liver, and a reduction of 55–65% in lung tissue upon tamoxifen treatment compared to vehicle treatment (Fig. EV1A–C). The reduction of GRα expression was already observed 1 week after

the last tamoxifen treatment, and it persisted for at least 4 weeks. As expected, no significant reduction in GRα expression was found in the brain upon tamoxifen treatment (Fig. EV1D), consistent with the notion that the used tamoxifen regimen does not easily cross the blood-brain barrier (Friedel et al, 2011). Efficient GR deletion was also confirmed in adrenal gland, pancreas, muscles, white adipose tissue and kidney extracts in tamoxifen-treated GRiKO 4 weeks after the final tamoxifen treatment (Fig. EV1E–I). Overall, we confirmed the successful GR deletion in most tissues of the GRiKO mice.

## Endogenous GCs ensure survival upon *Plasmodium chabaudi* AS infection

GRiKO and WT littermates were infected with *Pc*AS parasites to study the role of endogenous GCs in malaria tolerance. Mice were monitored daily, starting from 6 days post infection (dpi), as shown in Fig. 1A. Infection of C57BL/6J mice with *Pc*AS is known to be a mild non-lethal model of malaria whereby a transient parasitemia peak is observed (Van den Steen et al, 2010). Remarkably, approximately 50% of *Pc*AS-infected GRiKO mice succumbed to the disease between 10 and 13 dpi, while all WT mice survived (Fig. 1B). No difference in peak parasitemia was observed between both infected groups (Fig. 1C). Supporting this, no difference in systemic or liver parasite load was detected at 10 dpi (Appendix Fig. S1A,B). Mice were also scored for clinical disease severity. In both WT and GRiKO, clinical symptoms such as piloerection, trunk curl, reduced limb grasping and diminished body tone were detected upon infection. Yet, a significant increase in the severity of these symptoms was detected in infected GRiKO mice (Fig. 1D). Similar decreases in body weights were observed in both groups throughout the infection and were most pronounced shortly after the peak of parasitemia (Fig. 1E). In contrast, hypothermia (<35 °C; measured at 10 dpi) only occurred in infected GRiKO mice (Fig. 1F). We also measured liver and spleen weights at 10 dpi. In both infected groups, hepatosplenomegaly was detected, but GR deletion did not affect liver weight and only slightly increased spleen weight (Fig. 1G,H). Overall, these findings under-score the protective function of GR signaling in the context of experimental malaria.

## GR-deficient mice develop hypoglycemia during experimental malaria

Given the important glucose-regulating effects of the GR and the metabolic disturbances reported in malaria patients, we monitored glycemia levels daily from 6 dpi onwards (Madrid et al, 2015; Kuo et al, 2015). While glucose levels remained stable in uninfected mice and infected WT mice, infected GRiKO mice reaching humane endpoints during the infection exhibited severe hypoglycemia (defined as <40 mg/dL), particularly around 10 dpi (Fig. 1I,J; Appendix Fig. S1C–E). Concurrently, we assessed blood lactate levels, a key indicator of glycolytic activity. Elevated lactate levels were found in both infected groups, but further increased upon GR deletion (Fig. 1K). Hepatic glycogen, a major glucose reserve, was significantly depleted upon infection in both groups but was almost completely exhausted in GRiKO mice at 10 dpi (Fig. 1L). To further explore this glucometabolic dysregulation, we measured glucagon and insulin levels, two glucose-regulating hormones produced by the pancreas. Notably, infected GRiKO mice exhibited increased

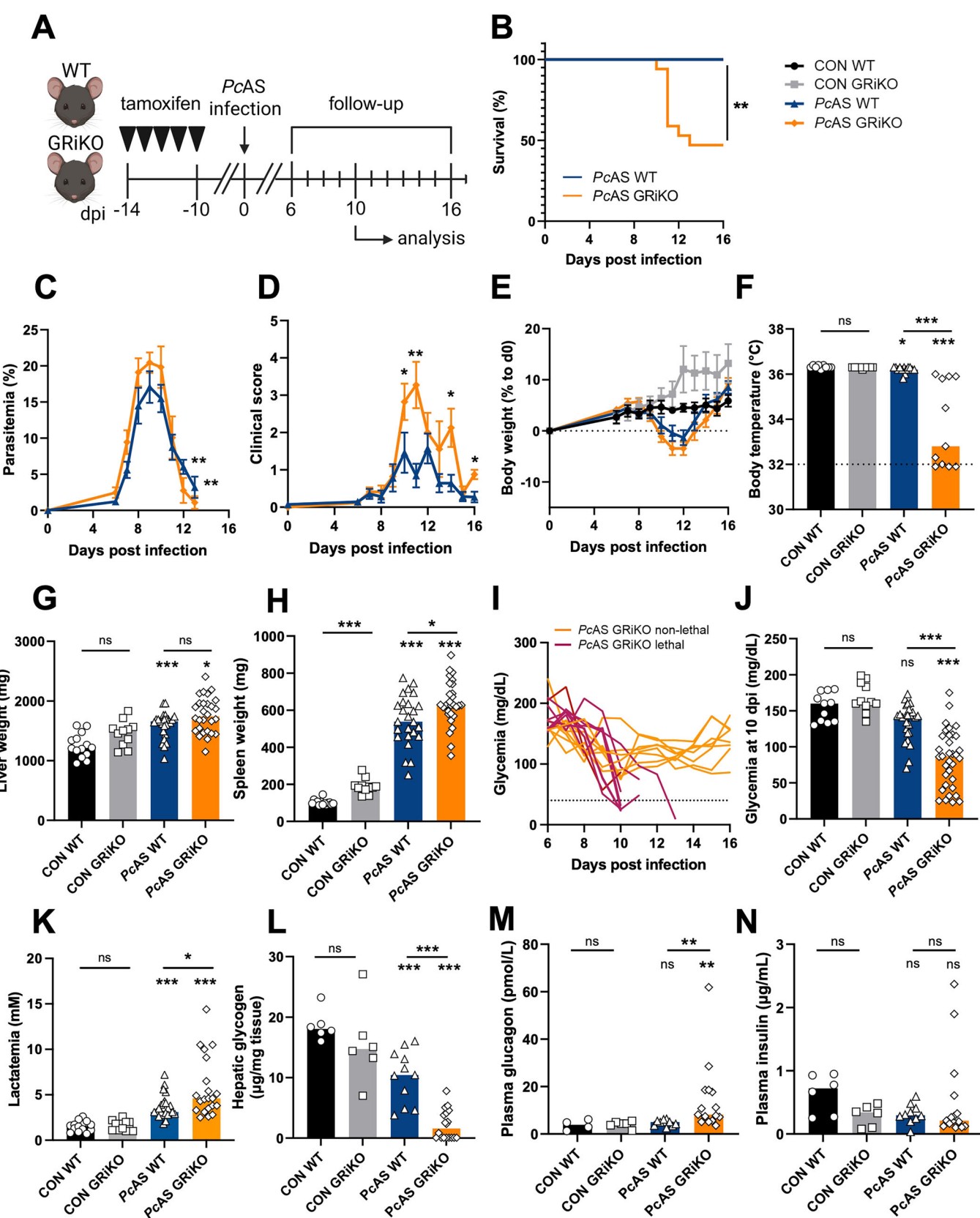

**Figure 1.  GRiKO mice develop lethal hypoglycemia upon *Pc*AS infection.**

WT and GRiKO mice were infected with $10^4$ *Pc*AS parasites and monitored from 6 until 16 dpi. **(A)** Schematic overview of experimental set-up is shown. **(B)** Survival rates, **(C)** parasitemia levels, **(D)** clinical scores and **(E)** body weight changes were analyzed daily. At 10 dpi, **(F)** body temperature was measured (dotted line: limit of detection (LOD). Body temperatures below the LOD were entered as 31.9 °C). **(G)** Liver and **(H)** spleen weights were measured at 10 dpi to assess hepatosplenomegaly. **(I)** Daily glycemia levels, measured from 6 dpi, are shown for *Pc*AS-infected GRiKO mice (red lines indicate lethal cases; dotted line indicates 40 mg/dL). **(J)** Glycemia levels measured at 10 dpi are shown, as well as **(K)** lactatemia, **(J)** hepatic glycogen stores, **(M)** plasma glucagon levels and **(N)** plasma insulin levels. For **(C–E)**, mean and SEM of each group is shown; **(F–N)** each datapoint or line represents an individual mouse. Statistical differences were calculated with the Mann–Whitney $U$ test with Holm–Bonferroni correction. Asterisks above data points indicate significant differences between infected and non-infected mice, asterisks above a horizontal line show significant differences between genotype. For all graphs: ns: $P > 0.05$, *$P < 0.05$, **$P < 0.01$, ***$P < 0.001$. Data from two or three independent experiments. The number of mice for graphs **(B–E)**: CON WT: 5 mice; CON GRiKO: 11 mice, *Pc*AS WT: 14 mice and *Pc*AS GRiKO: 17 mice. Source data are available online for this figure.

glucagon levels, while insulin levels remained unchanged (Fig. 1M,N), indicating that the observed hypoglycemia is insulin-independent.

Since not all infected GRiKO mice succumbed to the infection, we first investigated whether this variability in outcome might be attributed to incomplete GR deletion (Appendix Fig. S2). However, GR expression analysis in GRiKO mice that survived the full course of infection (up to 16 dpi) confirmed efficient and complete GR deletion, indicating that differential survival was not due to incomplete recombination. Hence, we further analyzed the lethal and non-lethal cases separately based on additional clinical parameters (Fig. EV2). Mice that died during infection displayed comparable peak parasitemia (Fig. EV2A–C) and body weight loss (Fig. EV2D–F) as surviving mice, but had significantly higher clinical scores (Fig. EV2G–I) and more profound hypoglycemia (Fig. EV2J–L). These findings highlight the association between hypoglycemia and mortality in GRiKO mice. Further analysis revealed a strong positive correlation between glycemia, body temperature, hepatic glycogen and hepatic parasite load (Fig. EV3A–C), while a negative correlation with glucagon levels and lactatemia was observed and no correlation with insulin levels. (Fig. EV3D–F). Overall, these results underscore the critical role of GR-mediated processes in regulating glucose metabolism during malaria, and show a clear association between hypoglycemia and lethality in the absence of GR signaling.

## FDG-PET/MRI reveals enhanced glucose retention in the liver and spleen of infected GRiKO mice

The severe metabolic changes seen in GRiKO mice during malaria infection led us to investigate in which specific organ(s) glucose is taken up and/or retained. Therefore, we performed in vivo $^{18}$F-fluorodeoxyglucose positron emission tomography magnetic resonance imaging ([$^{18}$F]-FDG-PET/MRI). For this imaging tool, [$^{18}$F]-FDG, a radiolabeled glucose analog, is injected into the mice and detected with a PET/MRI scanner after 1 h. In non-infected mice, [$^{18}$F]-FDG was primarily localized to the bladder due to its normal excretion pathway (Fig. 2A). However, in both infected WT and GRiKO mice, [$^{18}$F]-FDG accumulation was observed in the liver and spleen (Fig. 2A). To quantify this, organ-specific [$^{18}$F]-FDG uptake was calculated based on signal intensity measured directly from PET/MRI images. Remarkably, infected GRiKO mice exhibited a 3.8-fold increase in [$^{18}$F]-FDG signal per volume in the liver and a twofold increase in the spleen compared to infected WT mice (Fig. 2B,C). Moreover, in infected mice [$^{18}$F]-FDG uptake negatively correlated with glycemia levels, which were measured shortly before [$^{18}$F]-FDG administration (Fig. 2D,E). Overall, the

enhanced glucose uptake and/or retention in the liver and spleen of infected GRiKO mice, as visualized by [$^{18}$F]-FDG-PET/MRI, further supports the notion that GR-signaling is crucial for preventing excessive glucose utilization and metabolic dysregulation in the liver and the spleen.

## GR deletion minimally alters leukocyte numbers in the liver and spleen during malaria

To determine whether the increased [$^{18}$F]-FDG-PET/MRI signals in infected GRiKO mice were due to higher leukocyte numbers, we conducted a histological and flow cytometric analysis. Histological examination of liver tissue at 10 dpi showed significant leukocyte infiltration in the portal triad, along with sinusoidal dilation, hemozoin deposition, and disrupted liver architecture (Appendix Fig. S3). However, these alterations occurred regardless of GR presence and glycemic status. To further investigate leukocyte infiltration, we performed flow cytometry on liver and spleen tissues collected at 10 dpi (Appendix Figs. S4 and S5). Both organs exhibited substantial increases in lymphoid (CD4$^+$ T cells, CD8$^+$ T cells, B cells, NKT cells, NK cells) and myeloid populations (dendritic cells, neutrophils, eosinophils, Ly6c$^+$ and Ly6c$^-$ monocytes) following *Pc*AS infection. Crucially, GR deletion had only minimal effects on leukocyte infiltration. In the liver, slight increases in the number of CD4$^+$ T cells, CD8$^+$ T cells and eosinophils were noted (Appendix Fig. S4A,C,J), though the proportion of effector T cells (CD44$^+$ CD62L$^-$) remained unchanged (Appendix Fig. S4B,D). In the spleen, a modest reduction in Ly6C$^+$ monocytes was observed in infected GRiKO mice compared to WT mice (Appendix Fig. S5). These findings led us to conclude that the elevated [$^{18}$F]-FDG signal in the liver and spleen of GRiKO mice is not due to an increase in leukocyte numbers but likely reflects altered metabolic activity directly driven by GR deficiency.

## Cell-specific deletion of GR does not influence malaria tolerance

Given that aberrant glucose uptake and/or retention were mainly observed in the liver and the spleen of infected GRiKO mice, we hypothesized that GR-signaling is required in specific immune cells or in hepatocytes to maintain a balanced glucose metabolism during malaria. To test this hypothesis at the cellular level, we infected either hepatocyte-specific (GR$^{AlbCRE}$) (Fig. 3A–E), T-cell-specific (GR$^{LckCRE}$) (Fig. 3F–J) or myeloid cell-specific (GR$^{LysMCRE}$) (Fig. 3K–O) GR KO strains with *Pc*AS parasites. Disease progression and systemic glucose levels in these three cell-specific

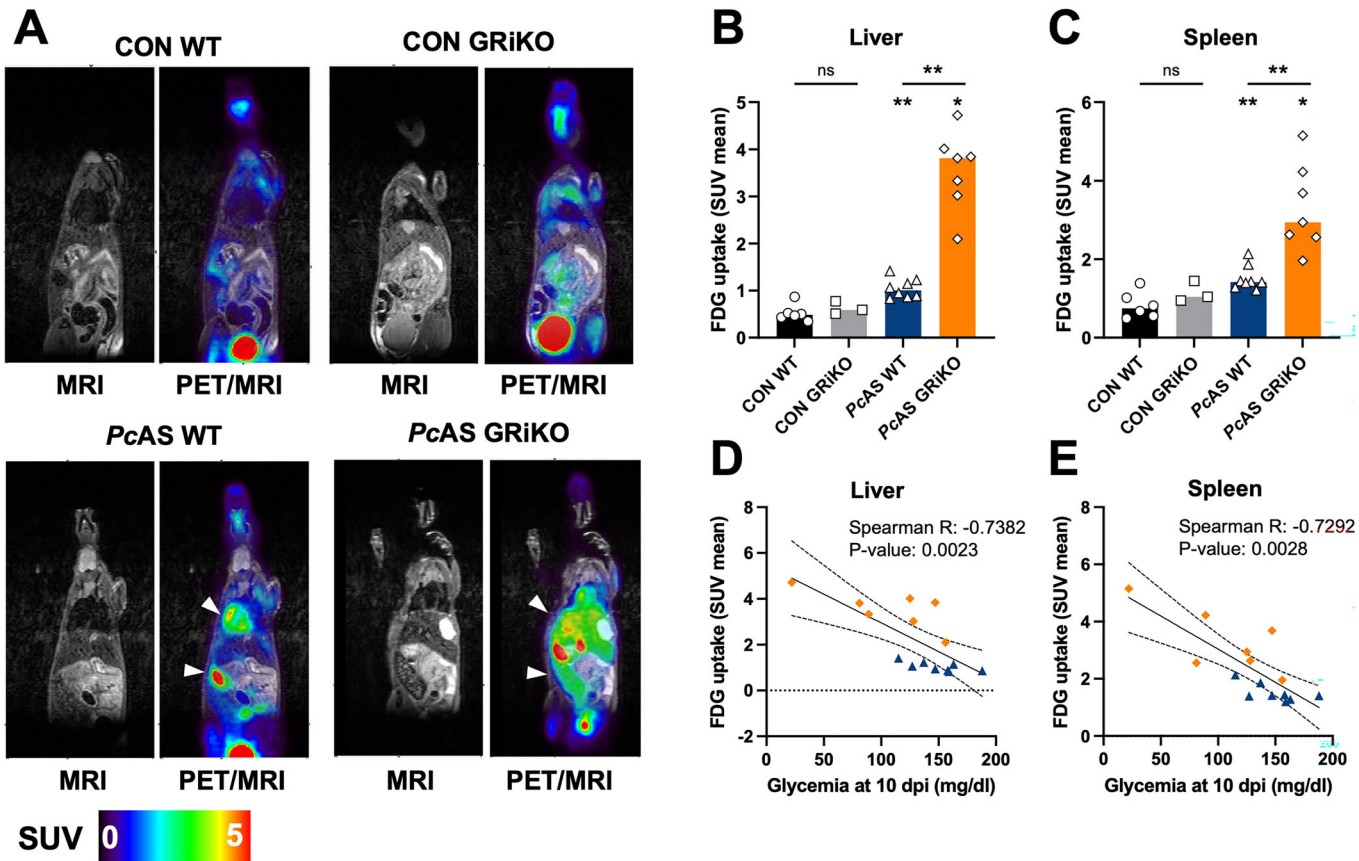

**Figure 2. [¹⁸F]-FDG-PET/MRI shows increased liver and spleen glucose retention in infected GRiKO mice.**

WT and GRiKO mice were infected with 10⁴ *Pc*AS parasites and [¹⁸F]-FDG-PET/MRI imaging was performed at 10 dpi. (A) One representative image of the [¹⁸F]-FDG-PET/MRI imaging per group is shown, with the dorsal anatomical plane depicted. For each condition, the left panel shows the MR imaging and the right panel shows the corresponding merged PET/MRI imaging. The arrows indicate the anatomical position of the liver (upper arrow) and spleen (bottom arrow). Mean standardized uptake values (SUV) in the (B) liver and (C) spleen were quantified. Each symbol represents an individual mouse. Correlation analysis between glycemia at 10 dpi and FDG uptake in the (D) liver and (E) spleen, is shown. (B, C) Statistical differences were calculated with the Mann–Whitney *U* test with Holm–Bonferroni correction. ns: *P* > 0.05, **P* < 0.05, ***P* < 0.01. Asterisks above data points indicate significant differences between infected and non-infected mice, asterisks above a horizontal line show significant differences between genotype. (D, E) Correlations were calculated with a nonparametric Spearman test. *P* values are represented on each graph. Each symbol represents an individual mouse with ▲: infected WT mice and ◆: infected GRiKO mice. For all graphs: Data from three independent experiments. Source data are available online for this figure.

KO mouse lines were closely monitored from 6 dpi onwards and compared to corresponding WTs. Surprisingly, none of the three cell-specific GR KO groups exhibited changes in survival, parasitemia levels or body weight during the course of infection, and most importantly, glycemia levels remained unaffected (Fig. 3E,J,O; Appendix Fig. S6). Notably, only the myeloid cell-specific GR knockout mice exhibited significant increases in clinical scores shortly after the peak of parasitemia (Fig. 3N). Overall, these results suggest that GR signaling across multiple cell types may be essential for inducing disease tolerance and managing glucose metabolism in *Pc*AS infection.

## GR signaling protects against upregulated glycolytic transcriptome during malaria

Given the systemic metabolic disturbances observed in infected GRiKO mice, along with the lack of significant effects from cell-specific GR knockout models, we investigated the role of GR signaling on the liver transcriptome during malaria infection. To achieve this, we performed RNA sequencing on whole liver tissue from uninfected and infected WT and GRiKO mice at 10 dpi (Dataset EV1). Principal component analysis (PCA) revealed a clear distinction between uninfected and infected mice (Appendix Fig. S7A), with livers from infected WT mice clustering separately from those of infected GRiKO mice. This finding already underscores the substantial influence of GR deletion on the liver transcriptome during infection. Differential expression gene (DEG) analysis showed minimal changes between the uninfected groups (Appendix Fig. S7B; Dataset EV1), but revealed dramatic transcriptomic alterations when comparing the livers of infected mice to their respective uninfected controls (Appendix Fig. S7C,D; Dataset EV1). Importantly, the loss of GR signaling during infection led to extensive changes in the liver transcriptome, identifying 819 DEGs when comparing infected WT to infected

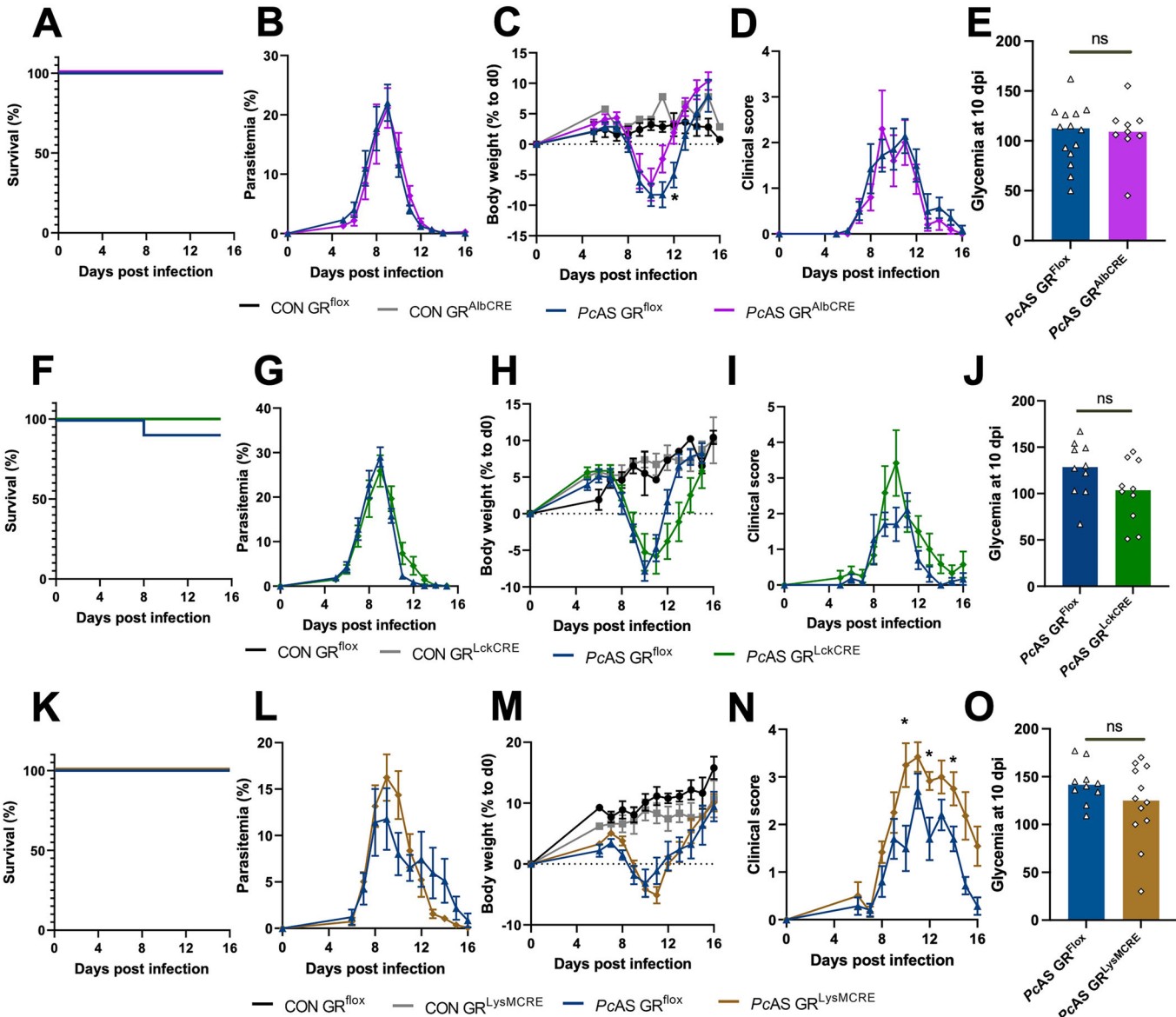

**Figure 3. Cell-specific deletion of GR does not influence malaria tolerance.**

Hepatocyte-specific (AlbCRE), T cell-specific (LckCRE) and myeloid-specific (LysMCRE) GR KO and GR^flox mice were infected with 10^4 PcAS parasites and monitored from 6 or 7 until 14 to 16 dpi. Survival rate, parasitemia, body weight loss, clinical score and glycemia at 10 dpi are shown for experiments performed with (A–E) AlbCRE mice, (F–J) LckCRE mice, and (F–O) LysMCRE mice. For each parameter, longitudinal data is represented via mean and SEM. For bar graph, each datapoint represents an individual mouse. Data from two or three individual experiments. For all graphs, statistical differences were calculated with the Mann–Whitney U test with Holm–Bonferroni correction. ns: non-significant, *P < 0.05. The number of mice for graphs (A–D): CON GR^Flox: 4 mice, CON GR^AlbCRE: 3 mice, PcAS GR^Flox: 9 mice and PcAS GR^AlbCRE: 14 mice, for graphs F-I: CON GR^Flox: 3 mice, CON GR^LckCRE: 3 mice, PcAS GR^Flox: 11 mice and PcAS GR^LckCRE: 11 mice, and for graphs (K–N): CON GR^Flox: 4 mice, CON GR^LysMCRE: 4 mice, PcAS GR^Flox: 10 mice and PcAS GR^LysMCRE: 12 mice. Source data are available online for this figure.

GRiKO mice (Fig. 4A; Dataset EV1). Of the 819 DEGs, 612 were upregulated and 207 were downregulated. The top upregulated genes included *Slc16a3* (a pyruvate/lactate transporter) and *Pfkp* (a glycolytic enzyme), while the most downregulated genes included *Ppp1r3b* (promoting glycogen synthesis) and *Slc37a4* (a glucose-6-phosphatase transporter) (Fig. 4A; Dataset EV1). To further assess the implications of these transcriptomic changes for glucose metabolism, we subclustered the 819 DEGs to identify those relevant to glucose-related pathways using three molecular pathway

databases: Reactome, Wikipathways and Gene Ontology. This analysis identified 33 DEGs associated with glucose metabolism. We visualized their expression on a heatmap, which revealed that most glucometabolic DEGs exhibited differential gene expression exclusively in the infected GRiKO group compared to all other groups (Fig. 4B).

To assess whether the expression of these genes directly correlates with glycemia levels, we confirmed the differential expression of 10 key glucometabolic genes, including glycolytic

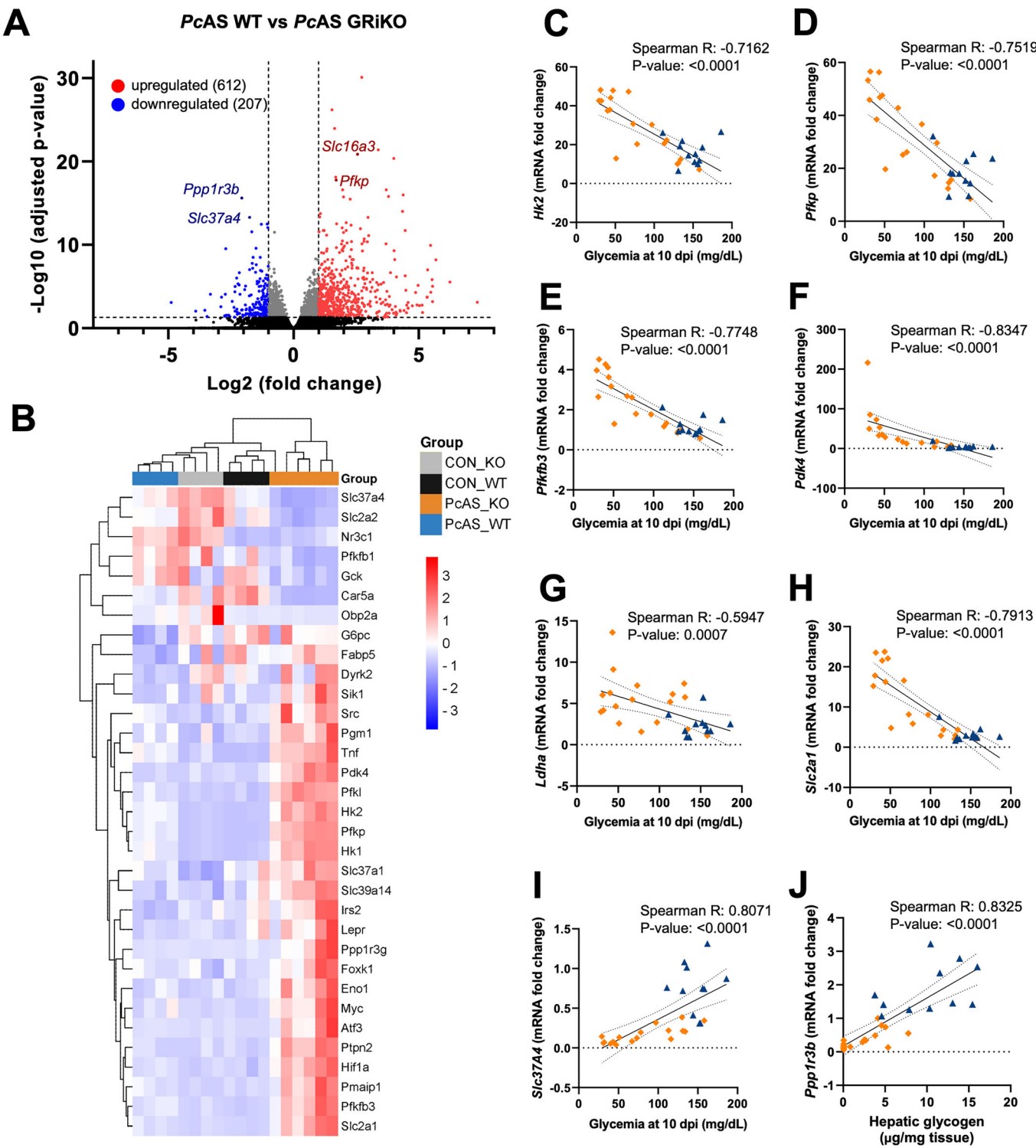

(*Hk2*, *Pfkp*, *Pfkfb3*, *Pdk4*, *Ldha*, and *Slc2a1*) and gluconeogenic genes (*G6pc*, *Pck*, *Ldhb*, and *Slc37a4*), with qPCR and conducted correlation analyses. We observed increased expression of the selected glycolytic genes, which negatively correlated with glycemia in both liver and spleen tissues (Figs. 4C–H and EV4; Appendix Fig. S8A–F), indicating that GR signaling suppresses excessive

glycolysis. In contrast, the expression levels of gluconeogenic genes *G6pc*, *Pck1*, and *Ldhb* did not show significant changes (Appendix Fig. S8G–I). However, *Slc37a4* was significantly downregulated in infected GRiKO mice and positively correlated with glycemia (Fig. 4I; Appendix Fig. S8J), suggesting that GR signaling primarily regulates gluconeogenesis at the level of glucose-6-phosphate

◀ **Figure 4.  GR signaling protects against upregulated liver-specific glycolytic transcriptome during malaria.**

WT and GRiKO mice were infected with $10^4$ *Pc*AS parasites and RNAseq was performed on liver tissue collected at 10 dpi. (A) A volcano plot was made to examine the differentially expressed genes (DEGs) comparing *Pc*AS-infected WTs with *Pc*AS-infected GRiKOs. DEGs discriminating *Pc*AS-infected WT from *Pc*AS-infected GRiKO were screened for their involvement in glucose metabolism. (B) A heatmap of the normalized counts of 33 identified glucometabolic genes is shown. Correlation analyses between glycemia levels and (C) *Hk2*, (D) *Pfkp*, (E) *Pfkfb3*, (F) *Pdk4*, (G) *Ldha*, (H) *Slc2a1*, and (I) *Slc37a4* are shown. (J) Correlation analysis between hepatic glycogen and *Ppp1r3b* is shown. (C–J) Correlations were calculated with a nonparametric Spearman test. *P* values are presented on each graph. Each symbol represents an individual mouse with ▲: infected WT mice and ◆: infected GRiKO mice. (A, B) Data from one independent experiment and graphs (C–J): data from two independent experiments. (CON WT: 4 mice, CON GRiKO: 4 mice, *Pc*AS WT: 4 mice, *Pc*AS GRiKO: 6 mice, with 3 normoglycemic and 3 hypoglycemic). Source data are available online for this figure.

transport rather than through enzyme expression. Furthermore, we confirmed the downregulation of *ppp1r3b* and detected a positive correlation between its expression and hepatic glycogen during infection (Appendix Fig. S8K; Fig. 4J), indicating that GR signaling helps to maintain glycogen levels. Overall, our findings demonstrate that GR signaling protects against an upregulated glycolytic transcriptome while promoting minimal gluconeogenesis during *Pc*AS infection, underscoring its vital role in maintaining metabolic balance in malaria.

## Increased JAK/STAT activation upon hypoglycemia in infected GRiKO mice

To explore new druggable pathways linked to lethal hypoglycemia, we used our RNAseq dataset (i.e, the 819 DEGs discriminating infected WT from infected GRiKO mice) to connect transcriptomic changes with transcription factor (TF) signaling (Fig. 5A). Using Chea3, a powerful TF enrichment analysis (TFEA) tool that uses publicly available chromatin immunoprecipitation sequencing (ChIP-Seq) databases, we identified Signal transducer and activator of transcription 3 (STAT3) as a key player (Keenan et al, 2019). STAT3 consistently ranked in the top 5 TFs across all three Chea3 databases (Fig. 5B–D), aligning with its known role in promoting glycolysis and suppressing hepatic gluconeogenesis (Inoue et al, 2004; Nie et al, 2009; Yucel et al, 2022; Camporeale et al, 2014). To validate this bioinformatic discovery, we experimentally measured STAT3 activation by assessing the ratio of phosphorylated STAT3 (pSTAT3, at tyrosine residue 705) to total STAT3. Our results revealed a marked increase in STAT3 activation in the liver upon *Pc*AS infection, which was further exacerbated in GRiKO mice (Fig. 5E). Strikingly, STAT3 activation strongly correlated with glycemia levels during infection, confirming its association with hypoglycemia (Fig. 5F). As a comparison, we also measured STAT1 activation, but its response to GR deletion was far less pronounced (Appendix Fig. S9), highlighting the specific involvement of STAT3 in glucose dysregulation. Additionally, we found higher expression of the upstream regulators *Il6* and *Il10* in GRiKO mice (Fig. 5G,H), both of which negatively correlated with glycemia (Fig. 5I,J). Similar results were obtained in the spleen (Fig. 6A–F). These findings suggest a pivotal link between the JAK/STAT3 pathway and glucose metabolism during *Pc*AS infection.

Furthermore, we also measured the mRNA expression levels of key cytokines i.e., *Il1β*, *Tnfα*, *Ifnα* and *Ifnγ*. In both liver and spleen, an increase in *Il1β* and *Tnfα* was detected in infected GRiKO mice compared to infected WT mice (Fig. EV5A,B,E,F), while *Ifnα* and *Ifnγ* were only increased in the liver (Fig. EV5C,D,G,H). This cytokine profile aligns with the well-

established anti-inflammatory role of GCs in modulating immune responses. Interestingly, the expression levels of *Il1β* and *Tnfα* in both liver and spleen correlated significantly with glycemia levels (Fig. EV5I,J,M,N), further underscoring the link between inflammation and glucose dysregulation. In addition, the liver-specific increase in *Ifnγ* also showed a significant correlation with glycemia (Fig. EV5K,L,O,P). Together, these findings support the notion that GC signaling serves as a critical modulator in preventing JAK/STAT pathway overactivation, cytokine storm and hypoglycemia during malaria.

## STAT3 enrichment in the whole blood transcriptome of severe malaria patients

To evaluate the clinical relevance of our findings, we analyzed a publicly available whole-blood RNAseq dataset from malaria patients. Since no transcriptomic datasets from hypoglycemic malaria patients are currently available, we evaluated a published dataset of pediatric malaria patients with either cerebral malaria and hyperlactatemia ("CH", as a proxy for dysmetabolism in severe malaria) versus uncomplicated malaria (accession number E-MTAB-6413) (Lee et al, 2018). A schematic overview of our computational analysis can be found in Fig. 7A. Pathway enrichment analysis using the WikiPathways database revealed a strong enrichment for aerobic glycolysis in CH patients, consistent with the metabolic alterations observed in infected GRiKO mice (Fig. 7B). Moreover, TFEA using the Chea3 platform further identified STAT3 as one of the top-ranked regulators of the differentially expressed genes (Fig. 7C), in line with our experimental findings of STAT3 hyperactivation upon severe disease in mice. In addition, signal transduction pathway analysis using the INOH database revealed significant enrichment of the JAK1 and JAK2 pathways, both which are critical upstream kinases that activate STAT3 through phosphorylation, in CH patients (Fig. 7D). These findings suggest that overactivation of the JAK/STAT3 signaling cascade may be a conserved feature of severe malaria pathogenesis in both murine and human malaria.

## Ruxolitinib treatment prevents the development of lethal hypoglycemia and hyperinflammation

Building on our discovery that JAK/STAT overactivation is associated with disease severity in murine and human malaria, we next evaluated whether pharmacological inhibition of JAK/STAT signaling with ruxolitinib could reverse the lethal phenotype in infected GRiKO mice. Ruxolitinib is a selective JAK1/2 inhibitor, which leads to inhibition of the activation of several STATs,

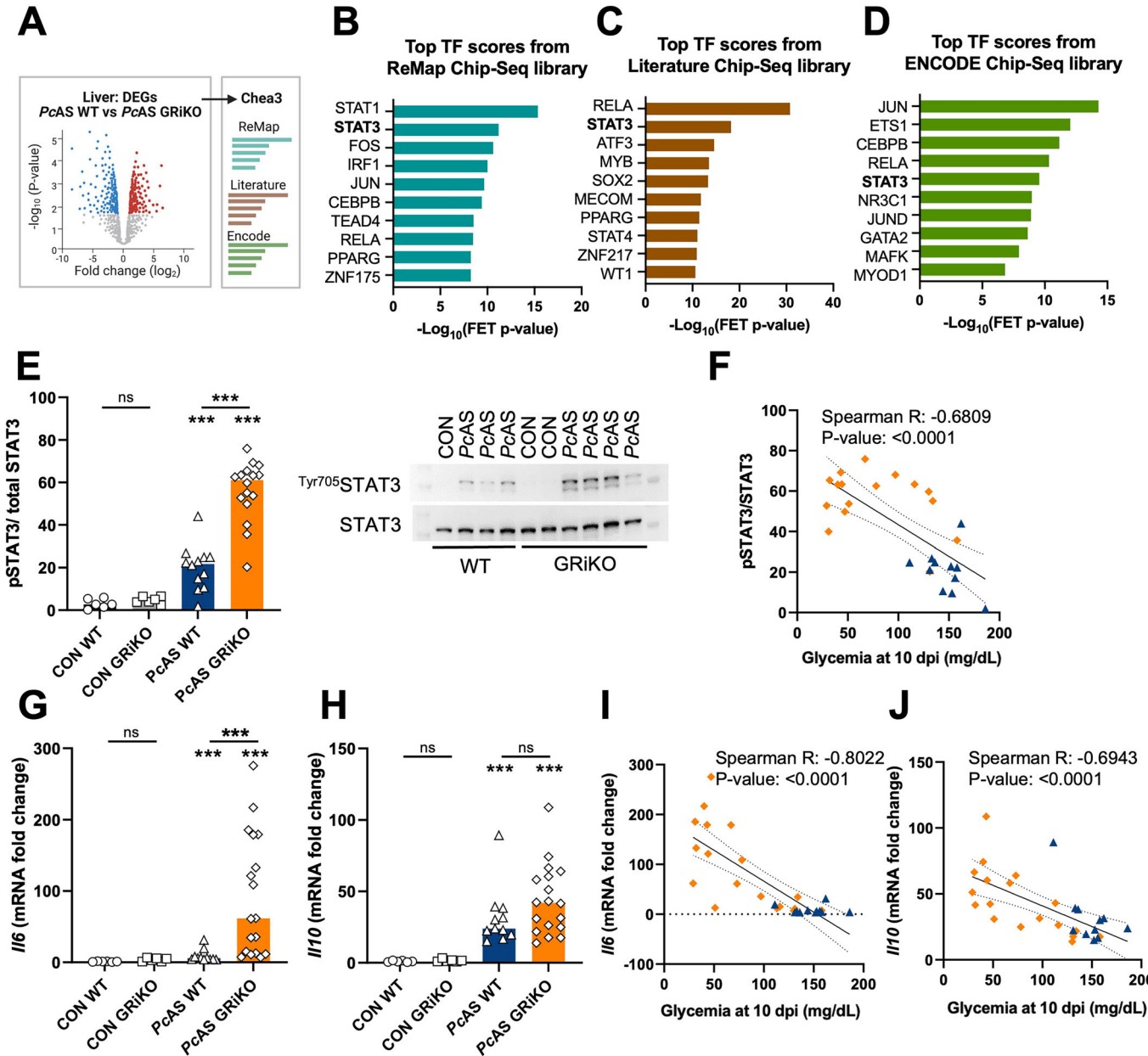

**Figure 5. Increased JAK/STAT activation upon hypoglycemia in liver of infected GRiKO mice.**

WT and GRiKO mice were infected with $10^4$ PcAS parasites and RNAseq was performed on liver tissue collected at 10 dpi. Differentially expressed genes (DEGs) discriminating PcAS-infected WT from PcAS-infected GRiKO mice, were identified with DESeq2. With the chea3 bioinformatic tool, transcription factor enrichment analysis (TFEA) was performed. (A) A schematic overview of the workflow is shown. TFEA results based on the (B) Remap, (C) Literature and (D) ENCODE public Chip-seq databases are shown (the position of STAT3 is marked in bold). (E) The phosphorylation (on tyrosine residue 705) status of STAT3 was assessed in liver tissue with western blot. For this analysis, (p)STAT3 was first detected, after which the blots were stripped and STAT3 was detected on the respective blots. Representative blots are shown together with the semiquantitative determination of band intensity ratio. (F) Correlation analysis of liver pSTAT3/STAT3 ratio with glycemia upon infection. mRNA expression of (G) Il6 and (H) Il10 were determined with qPCR. Correlation analysis of glycemia with (I) Il6 and (J) Il10 expression in liver tissue. (E, G, H) Statistical differences were calculated with the Mann–Whitney U test with Holm–Bonferroni correction. ns: $P > 0.05$, ***$P < 0.001$. Asterisks above data points indicate significant differences between infected and non-infected mice, asterisks above a horizontal line show significant differences between genotype. (F, I, J) Correlations were calculated using the nonparametric Spearman test. P values are presented on each graph. Each symbol represents an individual mouse with ▲: infected WT mice and ◆: infected GRiKO mice. For graphs (A–D): data from one independent experiment; For graphs (E–J): data from two independent experiments. Source data are available online for this figure.

including STAT3. We treated infected WT and GRiKO mice with ruxolitinib starting at 7 dpi (90 mg/kg, twice daily via oral gavage, Fig. 8A). This treatment regimen was chosen to mimic an early therapeutic intervention: initiated when the first clinical symptoms

were present, and with parasitemia already reaching ~5%. Strikingly, ruxolitinib significantly improved survival in GRiKO mice, with 88.9% of treated mice surviving, compared to just 38.5% in the vehicle-treated group (Fig. 8B). This marked improvement in

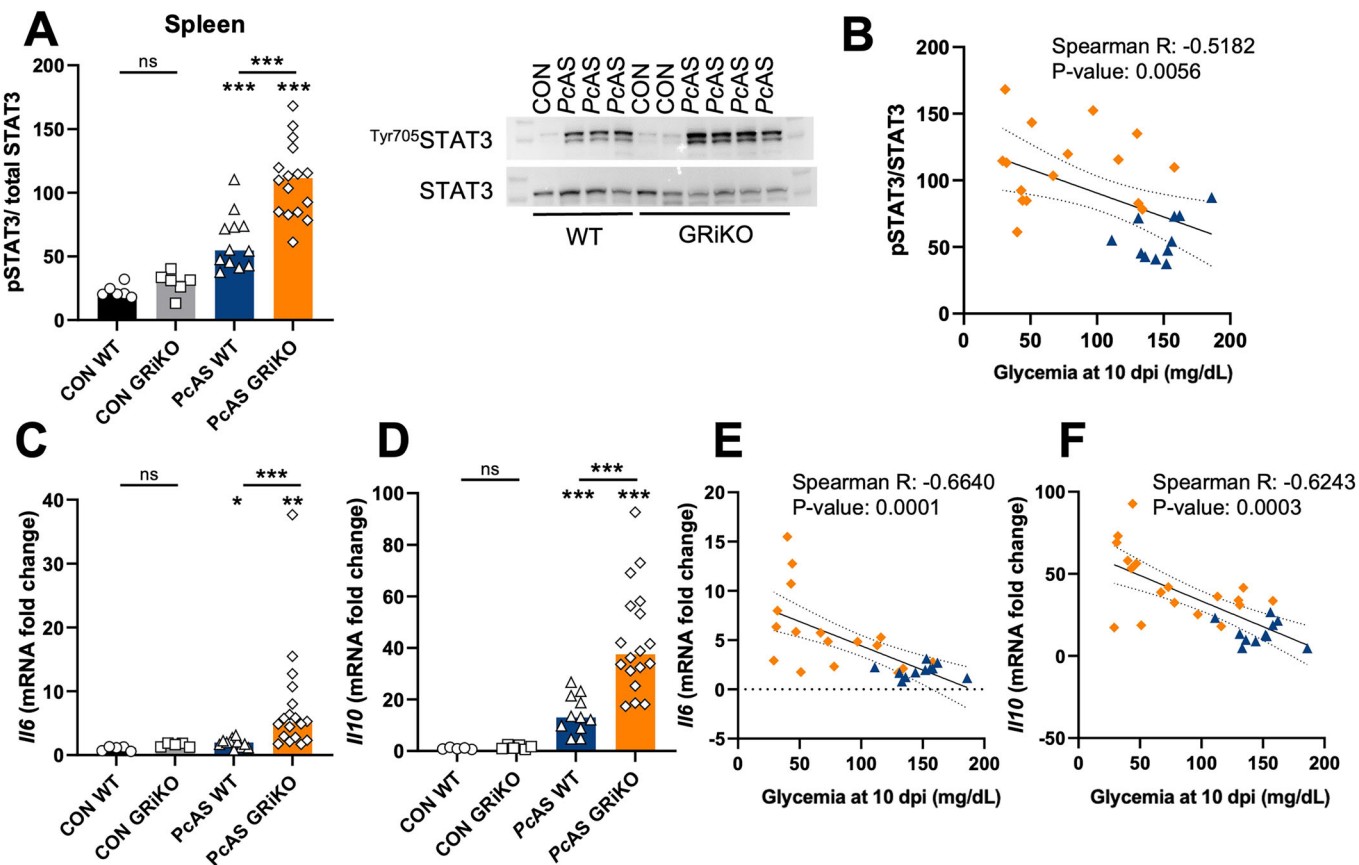

**Figure 6. JAK/STAT hyperactivation in the spleens of infected GRiKO mice.**

WT and GRiKO mice were infected with 10⁴ PcAS parasites. Spleen tissue was collected at 10 dpi. (A) The phosphorylation (on tyrosine residue 705) status of STAT3 was assessed in spleen tissue with western blot. For this analysis, (p)STAT3 was first detected, after which the blots were stripped and STAT3 was detected on the respective blots. Representative blots are shown together with the semiquantitative determination of band intensity ratio. (B) Correlation analysis of splenic pSTAT3/STAT3 ratio with glycemia upon infection. mRNA expression of (C) Il6 and (D) Il10 were determined with qPCR. Correlation analysis of glycemia with (E) Il6 and (F) Il10 expression in spleen tissue. (A, C, D) Statistical differences were calculated with the Mann–Whitney U test with Holm-Bonferroni correction. (A, C, D) ns: P > 0.05, *P < 0.05, **P < 0.01, ***P < 0.001. Asterisks above data points indicate significant differences between infected and non-infected mice, asterisks above a horizontal line show significant differences between genotype. (B, E, F) Correlations were calculated using the nonparametric Spearman test. P values are presented on each graph. Each symbol represents an individual mouse with ▲: infected WT mice and ◆: infected GRiKO mice. All graphs: Data from two independent experiments. Source data are available online for this figure.

survival demonstrates the critical role of JAK/STAT signaling in driving the fatal outcomes.

Ruxolitinib did not affect parasitemia, hepatic parasite load, clinical score, weight loss, lactatemia, or hepatic glycogen levels (Fig. 8C,D; Appendix Fig. S10A–D). Nevertheless, it effectively prevented severe hypoglycemia in nearly all treated GRiKO mice (Fig. 8E; Appendix Fig. S10E). This underscores that STAT3 overactivation is a key driver of lethal hypoglycemia. Importantly, ruxolitinib did not alter circulating glucagon or insulin levels, suggesting its protective effects are independent of classical glucose-regulating hormones (Appendix Fig. S10F,G). In addition to rescuing glycemia, ruxolitinib significantly inhibited the development of splenomegaly upon infection, in both WT and GRiKO mice (Fig. 8F). To further explore the metabolic impact of ruxolitinib, we assessed glycolytic and gluconeogenic gene expression and measured FDG uptake in liver and spleen. A significantly lower splenic Hk1 expression was observed in ruxolitinib-treated groups, suggesting suppression of glycolysis

(Fig. 8G). Nevertheless, FDG uptake and expression of other metabolic genes, including Pfkp, Pdk4, Slc2a1, G6pc, Slc37a4 and Pck1 remained unchanged upon ruxolitinib treatment of infected GRiKO mice (Appendix Fig. S11).

Western blot analysis confirmed the efficacy of ruxolitinib in inhibiting STAT3 activation in both liver and spleen, with a significant reduction in pSTAT3/total STAT3 ratios (Fig. 8H,I). Furthermore, ruxolitinib also markedly reduced circulating levels of IL-6 and IL-10 in infected GRiKO mice, while having no significant effect on IFN-γ and TNF-α (Fig. 9A–D). Correlation analysis revealed strong associations between glycemia and cytokine levels (Fig. 9E–H). Interestingly, hypoglycemic GRiKO mice had elevated plasma IL-6 and TNF-α levels compared to normoglycemic GRiKO mice (Fig. 9I–L). No such association was observed for IL-10 or IFN-γ. Together, these findings reveal the mechanistic link between JAK/STAT overactivation, systemic cytokine storm and severe hypoglycemia during malaria infection.

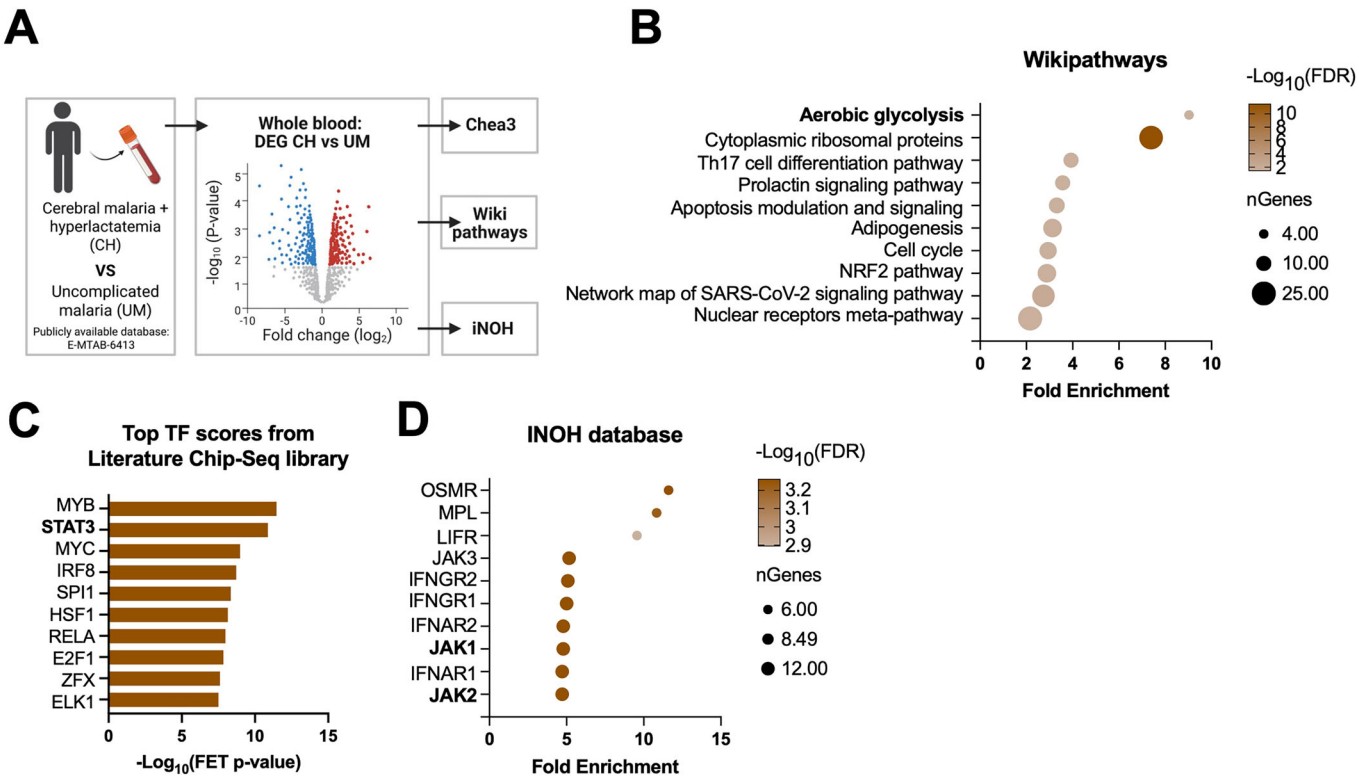

**Figure 7. JAK/STAT3 ranks among the top enriched pathways in severe malaria patients.**

Publicly available whole-blood RNAseq data from pediatric malaria patients (E-MTAB-6413, ArrayExpress database, EMBL-EBI) were analyzed to compare gene expression profiles between patients with cerebral malaria patients with hyperlactatemia (>5 mM; referred to as CH) and those with uncomplicated malaria (UM) (Lee et al, 2018). (A) Schematic overview of the computational workflow used for transcriptomic analysis. (B) Pathway enrichment analysis was performed using the WikiPathways database via the ShinyGO 0.80 platform. (C) Transcription factor enrichment analysis was conducted using the Literature-curated Chip-Seq database available in the Chea3 tool. The FDR-FET gene set enrichment analysis method was used. (D) Signal transduction pathway analysis was carried out using the INOH database, also via ShinyGO 0.80 tool. Differentially expressed genes (DEGs) between CH and UM groups were used as input for all analysis. Source data are available online for this figure.

## Ruxolitinib protects *Pb*NK65-infected mice

To evaluate the broader relevance of our findings, we also tested the therapeutic efficacy of ruxolitinib in an independent and clinically relevant model of severe malaria with C57BL/6 mice infected with *Pb*NK65-E parasites (Vandermosten et al, 2018b; Van den Steen et al, 2010). This model closely mimics key features of severe malaria in patients, including the development of hypoglycemia and GC resistance (Vandermosten et al, 2023, 2018a). Ruxolitinib was administered from 5 until 8 dpi (90 mg/kg, twice daily via oral gavage, Fig. 10A), corresponding to the early symptomatic phase. Ruxolitinib treatment abolished STAT3 activation in liver tissue of infected mice, confirming effective inhibition of the JAK/STAT3 pathway (Fig. 10B). Consistent with our observations in the *Pc*AS model, ruxolitinib did not affect parasitemia, but protected against severe symptoms, body weight loss and hypothermia (Fig. 10C–F). Most importantly, ruxolitinib also protected against hypoglycemia in this model, while no effect on lactatemia was detected (Fig. 10G,H). This corroborates our findings in the *Pc*AS-infected GRiKO mice. Collectively, these findings demonstrate that JAK/STAT3 inhibition with ruxolitinib effectively mitigates severe disease in two distinct experimental malaria models. This further

supports the potential of ruxolitinib as a promising adjunctive therapy for severe malaria.

## Discussion

Disease tolerance is a defense strategy that limits the fitness costs of infection without decreasing the pathogen burden. In this study, we reveal the pivotal role of GR signaling in mediating disease tolerance in malaria. Our previous findings already suggested a protective function for adrenal hormones in 4 different models of malaria, as evidenced by reduced survival rates of adrenalectomized mice upon infection (Vandermosten et al, 2018a). Furthermore, we documented GC resistance in a lethal experimental malaria model and diminished GC responses in leukocytes of severe malaria patients (Vandermosten et al, 2023). Within non-malaria infection models such as sepsis, GCs have also been recognized for their involvement in mediating disease tolerance (Vandewalle et al, 2021; Vettorazzi et al, 2015; Kleiman et al, 2012). However, mechanistic insights and therapeutic interventions to reverse lethal phenotypes remained lacking. Our study now identifies a GR–STAT3 axis that modulates systemic glucose

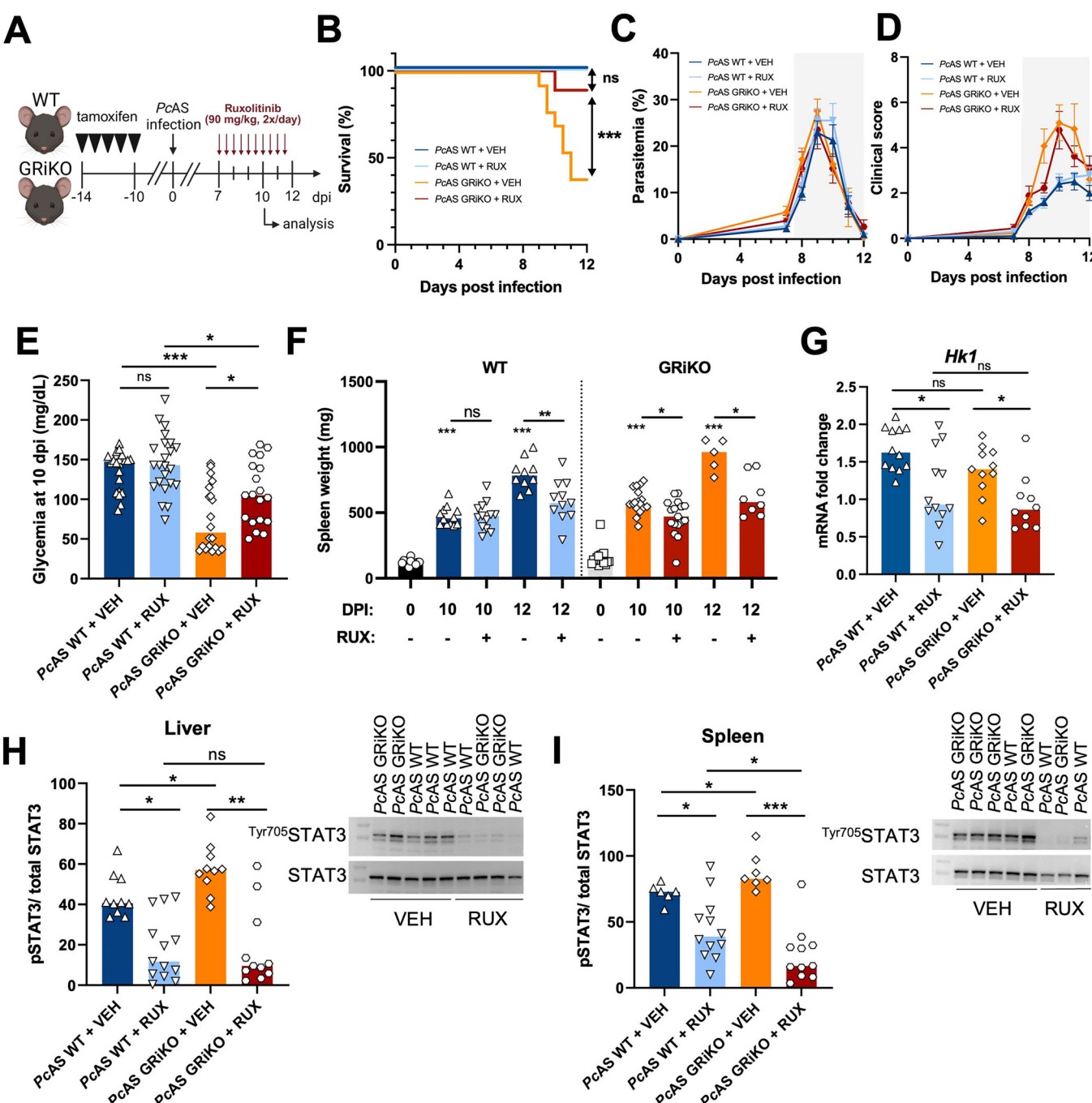

**Figure 8. Ruxolitinib treatment prevents the development of lethal hypoglycemia and hyperinflammation after PcAS infection.**

WT and GRiKO mice were infected with 10⁴ PcAS parasites. Mice were treated with 90 mg/kg ruxolitinib (RUX, JAK1/2 inhibitor) or vehicle (VEH) twice/day via oral gavage, from 7 dpi onwards. Mice were monitored from 6 dpi onwards. (A) Schematic overview of experimental set-up is shown. (B) Survival rates, (C) parasitemia levels and (D) clinical scores are shown. (E) Glycemia levels measured at 10 dpi. (F) Spleen weights were determined at 0, 10 and 12 dpi. (G) *Hexokinase 1* (*Hk1*) expression was measured in the spleen at 10 dpi. The ratio of phosphorylated STAT3 (pSTAT3, at tyrosine residue 705) over total STAT3 was assessed with western blot in (H) liver and (I) spleen extracts at 10 dpi. For this analysis, (p)STAT3 was first detected, after which the blots were stripped and STAT3 was detected on the respective blots. Representative blots are shown together with the semiquantitative determination of band intensity ratio. For graphs (C, D), mean and SEM of each group is shown. (E–I) Statistical differences were calculated with the Mann–Whitney *U* test with Holm–Bonferroni correction. Each datapoint represents an individual mouse. Asterisks above data points indicate significant differences between infected and their respective non-infected mice, asterisks above a horizontal line show significant differences between indicated groups. For all graphs: ns: $P > 0.05$, *$P < 0.05$, **$P < 0.01$, ***$P < 0.001$. Data from two to three independent experiments. The number of mice for graphs (B–D): CON WT: 3 mice, CON GRiKO: 3 mice, PcAS WT + VEH: 10 mice, PcAS WT + RUX: 10 mice, PcAS GRiKO + VEH: 13 mice, and PcAS GRiKO + RUX: 9 mice. Source data are available online for this figure.

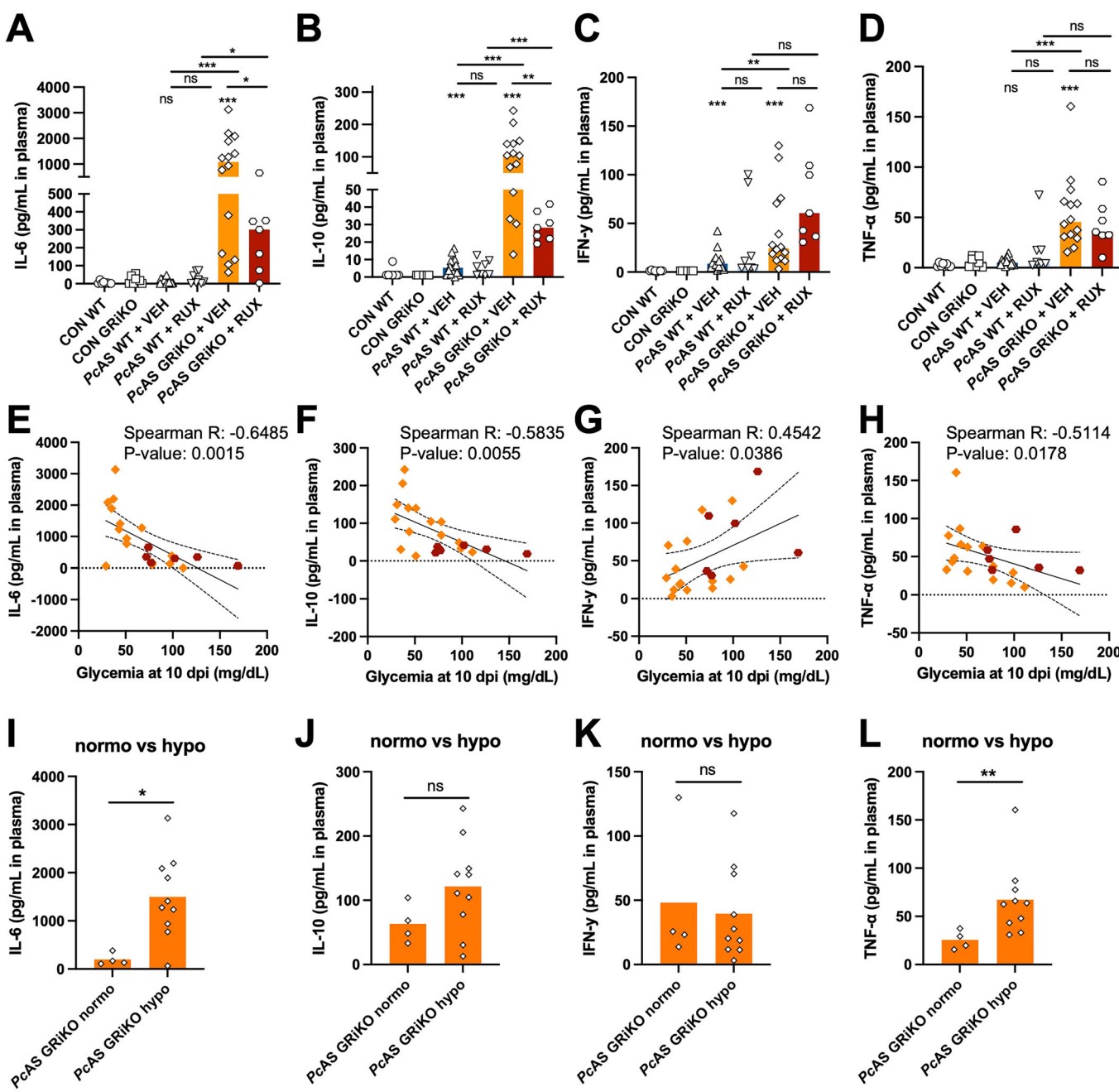

**Figure 9.  Ruxolitinib attenuates excessive systemic cytokine levels in infected GRiKO mice.**

WT and GRiKO mice were infected with 10⁴ PcAS parasites. Mice were treated with 90 mg/kg ruxolitinib (RUX, JAK1/2 inhibitor) or vehicle (VEH) twice/day via oral gavage, from 7 dpi onwards. At 10 dpi, plasma was collected and multiplex ELISA was performed to detect (A) IL-6, (B) IL-10, (C) IFN-γ, and (D) TNF-α levels. Correlation of glycemia with (E) IL-6, (F) IL-10, (G) IFN-γ and (H) TNF-α. (I–L) Cytokine levels were compared between infected vehicle-treated GRiKO mice with normoglycemia (normo) and severe hypoglycemia (hypo). (A–D, I–L) Statistical differences were calculated with the Mann–Whitney U test with Holm–Bonferroni correction. ns: P > 0.05, *P < 0.05, **P < 0.01, ***P < 0.001. Asterisks above data points indicate significant differences between infected and their respective non-infected mice, asterisks above a horizontal line show significant differences between indicated groups. (E–H) Correlations were calculated using the nonparametric Spearman test. P values are presented on each graph. ◆: infected GRiKO mice receiving with vehicle treatment and ⬢: infected GRiKO mice with ruxolitinib treatment. For all graphs, data from two independent experiments. Source data are available online for this figure.

homeostasis and cytokine responses, offering a new therapeutic opportunity.

GCs are important regulators of mammalian glucose homeostasis (Kuo et al, 2015). In patients, hypoglycemia is a defining feature of severe malaria and a frequent metabolic complication (Madrid et al, 2015). It is a key indicator of poor prognosis, especially in *Plasmodium*-infected children and pregnant women. Consistent with our earlier study showing lethal hypoglycemia in

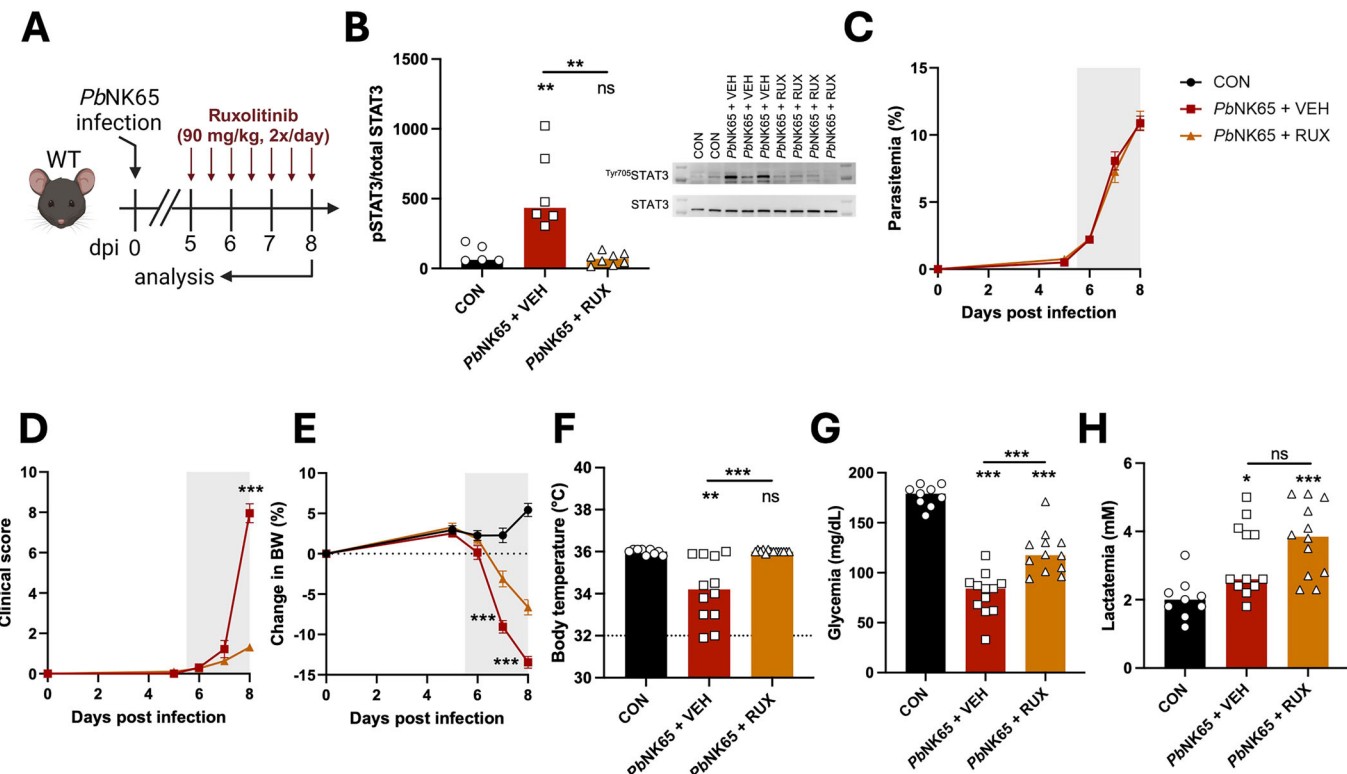

**Figure 10. Ruxolitinib protects *Pb*NK65-E-infected mice against severe disease.**

C57BL/6 mice were infected with $10^4$ *Pb*NK65-E parasites. Mice were treated with 90 mg/kg ruxolitinib (RUX) twice/day via oral gavage (or with vehicle, VEH), from 5 dpi until 8 dpi. Mice were monitored from 5 dpi onwards. (A) Schematic overview of the experimental set-up is shown. (B) At 8 dpi, liver tissues were collected to assess the ratio of phosphorylated STAT3 (pSTAT3, at tyrosine residue 705) over total STAT3 as a marker for STAT3 activation. For this analysis, (p)STAT3 was first detected, after which the blots were stripped and STAT3 was detected on the respective blots. Representative blots are shown together with the semiquantitative determination of band intensity ratio. (C) Parasitemia, (D) clinical score and (E) body weight loss were determined daily. (F) Body temperature, (G) glycemia levels and (H) lactatemia were measured on 8 dpi. Statistical differences were calculated with the Mann–Whitney $U$ test with Holm–Bonferroni correction. ns: $P > 0.05$, *$P < 0.05$, **$P < 0.01$, ***$P < 0.001$. (B, F–H) Each datapoint represents an individual mouse. Asterisks above data points indicate significant differences between infected and non-infected mice, asterisks above a horizontal line show significant differences between infected groups. For (C–E): mean and SEM of each group is shown. Asteriks indicate significant difference between infected groups. For all graphs, data from two independent experiments. Number of mice for graphs (B–D): CON: 13 mice, *Pb*NK65 + VEH: 23 mice, *Pb*NK65 + RUX: 19 mice. Source data are available online for this figure.

adrenalectomized *Plasmodium*-infected mice, we now demonstrate that GR deficiency leads to severe hypoglycemia upon *Pc*AS infection. In both patients and animal models, inducing similar levels of hypoglycemia e.g., through insulin overdose, can lead to functional brain failure and death, as seen in type 1 diabetes (Kaur and Seaquist, 2023; Cryer, 2007). Other studies support the importance of glucose metabolism in determining disease severity in malaria (Wang et al, 2018). Interestingly, GC resistance also increases the risk of developing hypoglycemia in sepsis, which suggests that our findings might also be of relevance for other diseases (Vandewalle et al, 2021).

Using [$^{18}$F]-FDG-PET/MRI, we detected that GR signaling restrains glucose uptake during infection, particularly in the liver and spleen. In other tissues and organs, no major increases in glucose uptake were detected, but because of incomplete GR deletion in the lungs, we cannot fully rule out additional protective effects in the lungs. Moderately increased [$^{18}$F]-FDG uptake in the spleen has also been detected during controlled human malaria infections (Woodford et al, 2021) and increased 2-(N-(7-Nitrobenz-2-oxa-1,3-diazol-4-yl)

Amino)-2-Deoxyglucose (2-NBDG, a fluorescent glucose analog) uptake has also been reported in monocytes and splenic T cells during human and murine malaria, respectively (Vandermosten et al, 2023; Miyakoda et al, 2018; de Salles et al, 2023).

Hypoglycemia can arise from either increased glucose consumption or impaired glucose production. GCs are critical to enhance hepatic expression of gluconeogenic genes (Kuo et al, 2015). Although we did not observe clear effects on *G6pc* and *Pck1* expression, we did observe a notable decrease in the expression of *Slc37a4* in the absence of GR. This gene encodes the glucose-6-phosphate transporter (G6PT) which facilitates the transport of glucose-6-phosphate from the cytoplasm into the lumen of the endoplasmic reticulum, where G6Pc metabolizes it to glucose and phosphate (Cappello et al, 2018). This suggests that GCs help maintain normal blood glucose levels, at least partially through supporting *Slc37a4* transcription. Consistent with our findings, GC regulation of the G6PT has previously been demonstrated in HepG2 and H4IIE cells (Hiraiwa H and Chou JY, 2001). Moreover, deficiencies in G6PT are also seen in glycogen storage disease type

1b, an autosomal recessive disorder marked by hypoglycemia and excessive glycogen accumulation in the liver (Hiraiwa H and Chou JY, 2001). However, in our study, a complete depletion of hepatic glycogen stores was detected in infected GRiKO mice, indicating the involvement of additional glucometabolic pathways. Indeed, we also observed reduced expression of *Ppp1r3b*, a gene encoding a phosphatase that activates glycogen synthase, indicating decreased glycogen production and increased glycogen breakdown upon GR deficiency. Mehta et al, demonstrated that liver-specific *Ppp1r3b* deletion in mice leads to decreased hepatic glycogen, lower plasma glucose and increased glycolytic enzymes expression in mice (Mehta et al, 2017). Consistent with this, we observed a significant increase in the expression of various glycolytic genes in *Pc*AS-infected GRiKO mice. GR-dependent regulation of *Ppp1r3b* expression in the liver was also detected in a study by Praestholm et al, in a mouse model of fasting-refeeding (Præstholm et al, 2021).

Despite the established role of GCs in suppressing insulin secretion, we did not detect any changes in insulin levels during infection or following GR deletion, indicating that severe hypoglycemia in infected GRiKO mice occurs independently of insulin. This aligns with findings in *Pc*AS-infected adrenalecto-mized mice, in which severe hypoglycemia also develops without significant changes in insulin, and in which treatment with clonidine, a well-known alpha-2 adrenergic agonist that inhibits insulin secretion from pancreatic beta cells, failed to prevent severe hypoglycemia (Vandermosten et al, 2018a). In addition, the profound depletion of glycogen stores in infected GRiKO and adrenalectomized mice, as well as reduced glycogen presence in post-mortem liver sections from malaria patients, further supports the notion of insulin-independent hypoglycemia (Vandermosten et al, 2018a; Brito et al, 1969). A few studies have reported hyperinsulinemia in malaria, however, these were induced by quinine, an anti-malarial drug that stimulates pancreatic beta cells to release insulin (Okitolonda et al, 1987; White et al, 1983). In fact, Kawo et al have shown that in severe falciparum malaria patients who did not receive quinine, hypoglycemia correlates with reduced plasma insulin levels compared to normoglycemic patients (Kawo et al, 1990). Thus, we uncovered a distinct severe glucose dysregulation in malaria infection, differing from viral infections like lymphocytic choriomeningitis virus (LCMV), where the relatively milder hypoglycemia is driven by IFN-γ-stimulated hyperinsulinemia and glycogen stores remain intact (Šestan et al, 2024).

Given the widespread expression of the GR, GCs have the potential to regulate various cell types. To explore whether GCs primarily maintain normal blood glucose levels by suppressing glucose consumption in immune cells, we utilized mice with specific deletions of the GR in T cells and myeloid cells. We also investigated whether the GR deficiency in the liver alone could trigger the development of hypoglycemia by employing mice with hepatocyte-specific GR deletions. Surprisingly, no lethality or hypoglycemia was detected during infection upon cell-specific GR deletion. This contrasts with the strong pro-inflammatory phenotypes observed in various disease models (Vettorazzi et al, 2015; Kleiman et al, 2012; Baschant et al, 2011). Although we cannot fully exclude the influence of incomplete cell-specific knockout in our mice, we postulate that GCs might balance gluconeogenesis and glycolysis during *Plasmodium* infection by

modulating crosstalk in multiple cell types or in a yet unknown cell type.

Besides severe hypoglycemia, infection of GRiKO mice was accompanied by markedly elevated cytokine levels, both systemically and within affected tissues, indicating that GC signaling also protects against a cytokine storm during malaria infection. This aligns with the well-documented anti-inflammatory effects of GCs (Cain and Cidlowski, 2017; De Bosscher and Haegeman, 2009), and with our previous findings in malaria-infected adrenalectomized mice (Van-dermosten et al, 2018a). A pronounced increase in a variety of cytokines is also a hallmark feature of severe malaria (Moxon et al, 2020; Wilairatana et al, 2022). Moreover, cytokine levels strongly correlated with glycemia upon infection, indicating a tight inter-connection between inflammatory and metabolic dysfunction. Con-sistent with this, our study also revealed an overactivation of the STAT3 pathway in the absence of GR signaling during infection. STAT3 hyperactivation has also been observed in other lethal experimental malaria models (Liu et al, 2013; Shi et al, 2008; Carpio et al, 2020; Chen et al, 2025), and IL-6, a key upstream activator of STAT3, is significantly elevated in severe malaria patients compared to those with non-severe malaria (Wilairatana et al, 2022). Consistent with findings in other conditions where STAT3 has been associated with reduced gluconeogenesis and increases glycolysis (Inoue et al, 2004; Nie et al, 2009; Yucel et al, 2022; Camporeale et al, 2014; Li M et al, 2017), we observed a strong inverse correlation between STAT3 activation and glycemia during *Plasmodium* infection. These findings further link immune dysregulation to metabolic complications. Importantly, our bioinformatic analysis of a published RNAseq dataset suggests that STAT3 activation is one of the hallmark features in severe malaria patients with metabolic dysfunction. This further emphasizes the translational relevance of our findings.

The current guidelines for treating severe malaria patients recommend the use of anti-malarial drugs, such as the artemisinin-based combination therapies (World Health Organization, 2023). Although these drugs are very efficient in killing the parasite, a significant number of patients still succumb to the disease, and there are currently no available adjuvant treatments (Varo et al, 2018). While our data confirm that GCs are protective, previous clinical trials using dexamethasone, a synthetic GC, showed no benefit in cerebral malaria (Warrell et al, 1982; Hoffman et al, 1988). This apparent discrepancy may be explained by GC resistance, which we have documented in both experimental models and patient samples (Vandermosten et al, 2023). With our current study, we propose ruxolitinib as a novel approach for treating severe malaria.

Ruxolitinib treatment ensured the survival of infected GRiKO mice by preventing both lethal hypoglycemia and an excessive cytokine storm. Its protective effect extended beyond GR-deficient models, as ruxolitinib also alleviated clinical symptoms and hypoglycemia in *Pb*NK65-E-infected WT mice. This underscores its relevance for a broader spectrum of severe malaria cases. Interestingly, ruxolitinib has previously been shown to elevate glucose levels in myeloproliferative neoplasms, which are clonal disorders of hematopoietic stem cells driven by gain-of-function mutations in e.g., JAK2 (Nageswara Rao et al, 2019). Despite these glucoprotective effects, the precise mechanisms by which ruxoliti-nib acts remain to be defined. Ruxolitinib did not alter FDG uptake and had minimal impact on the expression of glycolytic and gluconeogenic genes, aside from a suppression of *Hk1*. These observations suggest that ruxolitinib operates through alternative

or downstream pathways not captured by FDG-based assays, potentially involving post-translational regulation of glucose metabolism. Recent evidence shows that GCs also reprogram cellular glucose metabolism via non-genomic activities of the GR, raising the possibility that similar non-genomic mechanisms may be influenced by JAK/STAT3 signaling (Auger et al, 2024). Future studies exploring the non-genomic interplay between GR and STAT3 may reveal key regulatory nodes.

Notably, hypoglycemia in malaria patients is not only observed at hospital admission, but regularly also develops during anti-malarial treatment, where it is then strongly associated with poor outcomes (Chastang et al, 2023; Madrid et al, 2015). Interestingly, a recent study by Chen et al demonstrated that STAT3 remains active in the brains of *Pb*ANKA-infected mice even after 3 days of anti-malarial treatment, despite complete parasite clearance (Chen et al, 2025). These findings highlight the importance of host-directed therapies that address ongoing immunopathology beyond parasite clearance. Given the persistent activation of STAT3 and the emergence of hypoglycemia during treatment, it is highly important to investigate whether co-administration of ruxolitinib with anti-malarial drugs could improve survival in *Plasmodium*-infected mice and malaria patients. Currently, ruxolitinib is already Food and Drug Administration (FDA)-approved for treating myelofibrosis, polycythemia vera, steroid-refractory acute graft-versus-host disease and non-segmental vitiligo (Ostojic et al, 2011; Raedler LA, 2015; Martini et al, 2022; Sheikh et al, 2022). Moreover, a recent study already confirmed the safety and efficacy of ruxolitinib when co-administrated with anti-malarial drugs (Artemether-Lumefantrine) in healthy adults (Chughlay et al, 2022), supporting its feasibility as an adjunctive therapy in clinical malaria.

In conclusion, we reveal that endogenous GCs are critical components of disease tolerance during *Plasmodium* infection. Without reducing pathogen load, GR signaling suppresses JAK/STAT activation, and thereby protects against lethal insulin-independent hypoglycemia and an excessive cytokine storm. Moreover, we identify the therapeutic potential of ruxolitinib, a JAK1/2 inhibitor, in treating malaria-induced hypoglycemia and cytokine storm, advocating for its further investigation as a novel treatment for severe malaria.

## Methods

### Reagents and tools table

| Reagent/resource | Reference or source | Identifier or catalog number |
| --- | --- | --- |
| **Experimental models** | | |
| Nr3c1tm2GscGt(Rosa)26Sortm9(cre/ESR1)Arte mice | This study | N/A |
| Homozygous Nr3c1tm2Gsc mice | This study | N/A |
| Hemizygous Tg(Alb1-cre)7Gsc mice | This study | N/A |
| Hemizygous Tg(Lck-cre)1Cwi mice | This study | N/A |
| Hemizygous Lyz2tm1(cre)Ifo mice | This study | N/A |
| C57BL/6J | Janvier | N/A |
| **Antibodies** | | |
| CD3 – FITC (clone 145-2C11) | eBioscience | 11-0031-81 |

| Reagent/resource | Reference or source | Identifier or catalog number |
| --- | --- | --- |
| CD8 – PerCP-CY5.5 (clone 53-6.7) | eBioscience | 45-0081-82 |
| CD44 – PE (clone IM7) | Biolegend | 103007 |
| NK1.1 – PE-Cy7 (clone PK136) | eBioscience | 25-5941-81 |
| CD4 – APC e780 (clone RM4-5) | eBioscience | 47-0042-80 |
| B220 – BV786 (clone RA3-6B2) | BD | 563894 |
| CD62L – BV711 (clone MEL14) | Biolegend | 104445 |
| CD45 – BUV395 (clone 30-F11) | BD | 564279 |
| CD45 – FITC (clone 30-F11) | Biolegend | 103107 |
| CD11c – PE-Cy7 (clone N418) | Biolegend | 117318 |
| F4/80 – PE (clone BM8) | eBioscience | 12-4801-82 |
| SiglecF – PE-CF594 (clone E50-2440) | BD | 562757 |
| Ly6G – AF700 (clone 1A8) | BD | 561236 |
| Ly6C – APC-Cy7 (clone AL-21) | BD | 560596 |
| CD11b – eF450 (clone TM-25) | eBioscience | 48-0112-80 |
| MHCII – Horizon v500 (clone M5/114) | BD | 562366 |
| CD3 – BV650 (clone 17A2) | Biolegend | 100229 |
| CD19 – BV650 (clone 6D5) | Biolegend | 115541 |
| NK1.1 – BV650 (clone PK136) | Biolegend | 108736 |
| Rabbit anti-mouse GR | Cell Signaling Technologies | D8H2 |
| Rabbit anti-mouse Beta-actin | Cell Signaling Technologies | 4967 |
| Mouse anti-mouse HRP-tubulin | Thermo Fisher Scientific | BT7R |
| Rabbit anti-mouse vinculin | Cell Signaling Technologies | E1E9V |
| Rabbit anti-mouse STAT3 | Cell Signaling Technologies | 12640 |
| Rabbit anti-mouse STAT1 | Cell Signaling Technologies | 9172 |
| Rabbit anti-mouse phospho-STAT3 | Cell Signaling Technologies | 9145 |
| Rabbit anti-mouse phospho-STAT1 | Cell Signaling Technologies | 9167 |
| Donkey anti-rabbit HRP IgG (H + L) | Jackson ImmunoResearch | AB10015282 |
| **Oligonucleotides and other sequence-based reagents** | | |
| GR1 (GR) | This study | N/A |
| GR4 (GRflox) | This study | N/A |
| GR8 (GR wild-type) | This study | N/A |
| CRE1 | This study | N/A |
| CRE2 | This study | N/A |
| *18S* - mouse | Integrated DNA Technologies | Hs.PT.39a.22214856.g |
| *18S* – *Pc*AS parasite | Integrated DNA Technologies | N/A |
| *G6pc* | Integrated DNA Technologies | Mm.PT.58.11964858 |

| Reagent/resource | Reference or source | Identifier or catalog number |
| --- | --- | --- |
| Hk2 | Integrated DNA Technologies | Mm.PT.58.32698746 |
| Ldha | Integrated DNA Technologies | Mm.PT.58.29860774 |
| Ldhb | Integrated DNA Technologies | Mm.PT.58.9685691 |
| Pck1 | Integrated DNA Technologies | Mm.PT.58.11992693 |
| Pdk4 | Integrated DNA Technologies | Mm.PT.58.9453460 |
| Pfkfb3 | Integrated DNA Technologies | Mm.PT.58.29645117 |
| Pfkp | Integrated DNA Technologies | Mm.PT.58.23785960 |
| Ppp1r3b | Integrated DNA Technologies | Mm.PT.58.6871404 |
| Slc2a1 | Integrated DNA Technologies | Mm.PT.58.7590689 |
| Slc37a4 | Integrated DNA Technologies | Mm.PT.58.31385792 |
| Slc16a4 | Integrated DNA Technologies | Mm.PT.58.43799834 |
| Il1β | Integrated DNA Technologies | Mm.PT.58.41616450 |
| Il6 | Integrated DNA Technologies | Mm.PT.58.10005566 |
| Il10 | Integrated DNA Technologies | Mm.PT.58.13531087 |
| Tnf | Integrated DNA Technologies | Mm.PT.58.12575861 |
| Ifnα | Integrated DNA Technologies | Mm.PT.58.43426930 |
| Ifnγ | Integrated DNA Technologies | Mm.PT.58.41769240 |
| **Chemicals, enzymes, and other reagents** | | |
| 4-Aminobenzoic acid sodium salt | Sigma-Aldrich | A6928 |
| Tamoxifen | Sigma-Aldrich | T2859 |
| EZNA tissue DNA kit | Omega BioTek | D3396-00S |
| Giemsa | VWR | 1.09204.1000 |
| Isoflurane | Dechra | 2909737 |
| Ruxolitinib phosphate | MedChemExpress | HY-508588 |
| Dolethal | Vétoquinol | 1400522 |
| 4% paraformaldehyde | TissuePro Technology | PFA04-32R |
| Mouse Insulin ELISA | Mercodia | 10-1247-01 |
| Animal Glucagon ELISA | Mercodia | 10-1281-01 |
| Glycogen assay kit | Sigma-Aldrich | MAK016 |
| ACK lysing buffer | Gibco | A1049201 |
| Zombie Aqua Fixable Viability kit | Biolegend | 423102 |
| Zombie UV Fixable Viability kit | Biolegend | 423108 |

| Reagent/resource | Reference or source | Identifier or catalog number |
| --- | --- | --- |
| RNeasy Mini Kit | Qiagen | 74106 |
| High-Capacity cDNA Reverse Transcription kit | Applied Biosystems | 4368814 |
| TaqMan Fast Universal PCR master mix | Applied Biosystems | 4352042 |
| QuantSeq 3'mRNA-Seq Library Prep Kit | Lexogen | 015 |
| Phosphatase inhibitor cocktail 2 | Sigma-Aldrich | P5726 |
| Phosphatase inhibitor cocktail 3 | Sigma-Aldrich | P0044 |
| Protease inhibitor | Roche | 11836170001 |
| Pierce BCA Protein Assay Kit | Thermo Fisher Scientific | 10741395 |
| 10% Tris-Glycin Novex WedgeWell | Thermo Fisher Scientific | XP00102BOX |
| 4-12% Tris-Glycin Novex WedgeWell | Thermo Fisher Scientific | XP04122BOX |
| Bovine serum albumin | Carl Roth | 8076.3 |
| Non-fat dry milk | BioRad | 1706404 |
| SuperSignal West Pico PLUS chemiluminescent substrate | Thermo Fisher Scientific | 34580 |
| SuperSignal West Femto Maximum Sensitivity Substrate | Thermo Fisher Scientific | 34094 |
| InstantBlue® Coomassie Protein Stain | Abcam | Ab119211 |
| **Software** | | |
| Graphpad PRISM (version 9.5.1) | Graphpad | N/A |
| FlowJo (version 10) | BD | N/A |
| LAS (version 4.2) | Leica Microsystems | N/A |
| ImageJ (version 8.0_172) | ImageJ | N/A |
| R studio | R studio | N/A |
| PMOD | PMOD Technologies | N/A |
| **Other** | | |
| OneTouch Verio glucometer | LifeScan | N/A |
| Lactate Plus meter | Nova Biomedical | N/A |
| iHealth PT3 infra-red thermometer | iHealth | N/A |
| Bruker Biospec 70/30 7 Tesla MR scanner with Albira Si PET insert | Bruker Biospin | N/A |
| IBA 18/9 cyclotron | Ion Beam Application | N/A |
| GentleMACS Dissociator | Miltenyi Biotech | N/A |
| BD LSR Fortessa | BD | N/A |
| Excelsior MS tissue processor | Thermo Fisher Scientific | N/A |
| HistoStar Workstation | Thermo Fisher Scientific | N/A |
| Microm HM 355S microtome | Thermo Fisher Scientific | N/A |

| Reagent/resource | Reference or source | Identifier or catalog number |
|---|---|---|
| Leica DM 2000 light microscope | Leica Microsystems | N/A |
| ABI 7500 Real-time PCR System | Applied Biosystems | N/A |
| Precellys 24 tissue homogenizer | Bertin Technologies | N/A |
| HiSeq 4000 sequencer | Illumina | N/A |
| CLARIOSTAR plus plate reader | BMG Labtech | N/A |
| Fusion Solo S | Vilber Lourmat | N/A |

## Methods and protocols

### Ethical statement

All experiments at the Rega Institute for Medical Research (KU Leuven, Belgium) were performed according to the regulations of the European Union (directive 2010/63/EU) and the Belgian Royal Decree of 29 May 2013, and were approved by the Animal Ethics Committee of the KU Leuven (License LA1210186, project P154/2016, P174/2024 and P220/2018, Belgium). All experiments at the University of Ulm were approved by the federal authorities for animal research of the Regierungspräsidium Tübingen, Baden-Wuerttemberg, Germany (approved animal experimentation number: 1446) and performed in adherence with the National Institutes of Health Guidelines on the Use of Laboratory Animals and the European Union Directive 2010/63/EU on the protection of animals used for scientific purposes.

### Mice

In all experiments, infections were performed in male and female mice of 7–10 weeks old. Each experimental control group contained similar numbers of each sex. All mice were housed in a specific pathogen-free (SPF) animal house, in individually ventilated cages with up to five mice per cage and received ad libitum high-energy food (Ssniff Spezialdiäte GMBH). Water was supplemented with 0.422 mg/ml 4-amino-benzoic acid sodium (PABA; Sigma-Aldrich) after infection to facilitate parasite growth.

The GRiKO mice were originally generated by cross-breeding homozygous Nr3c1[tm2Gsc] mice (often referred to as GR[flox/flox]) with hemizygous Gt(ROSA)26Sor[tm9(cre/ESR1)Arte] mice to obtain Nr3c1[tm2Gsc]Gt(Rosa)26Sor[tm9(cre/ESR1)Arte] mice (hereafter referred to as GR WT when Cre-negative and GRiKO when Cre positive) (Rapp et al, 2018; Tronche et al, 1999). The global GRiKO mice were bred in the SPF animal house of the Rega Institute for Medical Research, KU Leuven. When 5–6-week old, GR WT and GRiKO littermates were injected intraperitoneally (i.p.) for 5 consecutive days with 10 mg/kg tamoxifen (Sigma-Aldrich) dissolved in Ethanol:Sunflower oil (1:8). The percentage of C57BL/6 background of WT and GRiKO mice was characterized on genomic tail DNA from 3 GRiKO mice and 3 of their WT littermates via a genome-wide SNP analysis (Taconic Biosciences). The characterization of all SNPs can be found in Dataset EV2. Less than 2,68% of SNPs showed variation between the 6 analyzed mice. Variations in these specific SNPs were present in both Cre⁻ and Cre⁺ mice. Thus, the observed variation is not affecting the comparison between GR

WT and GRiKO mice, since in all experiments, littermates were used.

Mice with a cell-specific knockout of the GR under the control of the albumin (Alb), lymphocyte protein tyrosine kinase (Lck) or lysozyme M (LysM) promotor were generated by cross-breeding homozygous Nr3c1[tm2Gsc] mice with either hemizygous Tg[(Alb1-cre)7Gsc], hemizygous Tg[(Lck-cre)1Cwi] or hemizygous Lyz2[tm1(cre)Ifo] to, respectively, generate GR[AlbCRE], GR[LckCRE], GR[LysMCRE] KO mice (Hua et al, 1994; Clausen et al, 1999; Mueller et al, 2011; Kellendonk et al, 2000; Tuckermann et al, 2007; Baschant et al, 2011). All lines were congenic to C57BL/6, and littermates were used in all experiments. For the experiments where mice are injected with *Pb*NK65-E parasites, 7-8-week-old C57BL/6J mice were purchased from Janvier Labs.

### Validation of mouse genotype via PCR

The tamoxifen-induced recombination of the loxP locus of the GR was measured by PCR in ear, tail, spleen, liver and/or brain tissue. DNA was isolated using the EZNA Tissue DNA Kit (Omega BioTek). PCR was performed with three primers for GR gene detection and two primers that detect the Cre recombinase gene (Appendix Table S1).

### Infection and clinical scoring

Infection was by i.p. injection of $10^4$ erythrocytes infected with the *Plasmodium chabaudi* AS (*Pc*AS) strain or the *Plasmodium berghei* NK65 (Edinburgh;*Pb*NK65-E). Both parasites were obtained from the late Prof. D. Walliker (Edinburgh University) (Van den Steen et al, 2010). Clinical parameters including social activity (SA), limb grasping (LG), body tone (BT), trunk curl (TC), pilo-erection (PE), shivering (Sh), abnormal breathing (AB), dehydration (D), incontinence (I) and paralysis (P) were evaluated daily to calculate a total clinical score of disease severity, as previously described (Pollenus et al, 2021). In brief, a disease score was given of 0 (absent) or 1 (present) for TC, PE, Sh and AB and 0 (normal), 1 (intermediate) or 2 (most serious) for the other parameters. The total clinical score was calculated by the following formula: $SA + LG + BT + TC + PE + 3*(Sh + AB + D + I + P)$. Body weight and body temperature were measured as well at the indicated time points. Mice were euthanized when humane endpoints were reached (clinical score of ≥10 or body weight loss of ≥20%) or at indicated time points.

### Determination of parasitemia, glycemia, lactatemia and anemia

Peripheral parasitemia (percentage of infected RBCs) was determined by microscopic analysis of blood smears after Giemsa staining (VWR). Blood glucose and lactate levels were measured in tail blood with the use of a OneTouch Verio glucometer (LifeScan) and a Lactate Plus meter (Nova Biomedical), respectively. Red blood cell concentration in the tail blood was determined in a Bürker chamber.

### In vivo [¹⁸F] fluorodeoxyglucose (FDG) uptake

[¹⁸F]-FDG uptake in vivo assessment with simultaneous positron emission tomography (PET) – Magnetic resonance imaging (MRI): The [¹⁸F]-FDG-PET radiotracer was prepared using an IBA 18/9 cyclotron and [¹⁸F]-FDG-PET synthesis module (Ion Beam Applications). At 10 dpi, mice were injected i.p. with $6.44 \pm 0.39$ MBq [¹⁸F]-FDG in 0.9% saline solution. After 45 min of uptake,

mice were anaesthetized with 2% isoflurane (Iso-Vet, Dechra) and scanning was initiated 15 min later. Simultaneous [$^{18}$F]-FDG-PET/MRI was performed with a Bruker Biospec 70/30 7 Tesla MR scanner equipped with an Albira Si PET insert (Bruker Biospin). Reconstructed PET images, corrected for decay, were simultaneously acquired and registered with MR images and normalized to injected radioactivity and animal weight to calculate standardized uptake values (SUVs). T$_2$-weighted MR images were acquired as previously described (Belderbos et al, 2020). A volume-of-interest (VOI)-based analysis was performed with the PMOD software (PMOD technologies). Spleen VOIs were outlined manually based on the MR images. In the liver, one VOI sphere was placed over the right lobe and one over de median lobe.

[$^{18}$F]-FDG uptake ex vivo assessment with automated gamma counter: At 10 dpi, mice were injected i.p. with $6.44 \pm 0.39$ MBq [$^{18}$F]-FDG in 0.9% saline solution. After 45 min, mice were anaesthetized with 2% isoflurane (Iso-Vet, Dechra) and sacrificed at specified timepoints for liver and spleen [$^{18}$F]-FDG uptake quantification. Liver and spleen were collected in tared tubes and weighed. Quantification of radioactivity in these organs was performed using an automated gamma counter equipped with a 3-inch NaI(Tl) well crystal coupled to a multichannel analyzer, mounted in a sample changer (Perkin Elmer 1480 Wizard 3q). Counts were corrected for background radiation, physical decay and counter dead time. The values have been expressed in percentage of the injected dose per organ. (%ID; organ activity (MBq)/injected activity (MBq)).

### In vivo JAK/STAT inhibitor treatment

For in vivo inhibition of JAK/STAT signaling, ruxolitinib phosphate (#HY-508588, MedChemExpress), a JAK1/2 inhibitor, was used. The inhibitor was dissolved in 30% PEG300 and 70% MilliQ (MQ) water (pH: 6.2). Ruxolitinib phosphate (90 mg/kg) was administered to *Pc*AS-infected mice twice a day, via oral gavage, from 7 dpi onwards, and to *Pb*NK65-infected mice from 5 dpi onwards.

### Collection of tissues

Mice were euthanized by i.p. injection of 100 µl of dolethal (Vétoquinol; 200 mg/ml). Blood samples were obtained by cardiac puncture with heparinized (LEO) syringes. Mice were systemically perfused (transcardial route) with 20 ml PBS to remove circulating blood. Organs were collected, weighed, and stored at −80 °C until further analysis. Livers were fixed in 4% paraformaldehyde (TissuePro Technology) for 48 h and stored in PBS for histological analysis. Spleens and livers were further processed for flow cytometry.

### Plasma insulin and glucagon determination

Blood samples were centrifuged and stored at −20 °C. Plasma insulin and glucagon levels were determined using a Mouse Insulin ELISA (#10-1247-01, Mercodia) and an Animal Glucagon ELISA (#10-1281-01, Mercodia), according to the manufacturer's protocol.

### Multiplex ELISA plasma

Plasma levels of IL-6, IL-10, IFN-γ, and TNF-α were determined with a custom-designed ProcartaPlex Multiplex immunoassay (Thermofisher), according to the manufacturer's protocol.

### Hepatic glycogen measurement

In total, 10 mg of each frozen liver sample was homogenized in 100 µl MQ using the Precellys 24 tissue homogenizer (Bertin Technologies). Then, liver homogenates were boiled for 5 min at 96 °C, centrifuged ($13,000 \times g$, 10 min, 4 °C) and supernatants was collected. To determine hepatic glycogen stores, a colorimetric Glycogen assay kit (#MAK016, Sigma) was used to determine hepatic glycogen stores, according to the manufacturer's protocol.

### Leukocyte isolation from liver tissue

During dissection, livers were collected in RPMI 1640 (Biowest) + 2% fetal calf serum (FCS) at room temperature (RT). Tissues were dissociated with the GentleMACS Dissociator (Miltenyi Biotec) and mashed through a 70-µm nylon cell strainer (VWR) to obtain single cells. After washing with PBS, pellets were dissolved in 37.5% Percoll and centrifuged ($700 \times g$, 12 min, at RT). After centrifugation, hepatocytes present in the top layer were first removed and RBC lysis was performed with ACK lysing buffer (Gibco) on the remaining cells (non-parenchymal cells). After the cells were washed with PBS, cells were resuspended in PBS + 2% FCS and live cells were counted in a Bürker chamber with trypan blue (VWR).

### Cell isolation from spleen tissue

During dissection, spleens were collected in PBS + 2% FCS and kept on 4 °C. Single cells were obtained after mashing the spleen through a 70-µm nylon cell strainer (VWR) followed by treatment with RBC lysis buffer (0.83% ammonium chloride (NH$_4$Cl; Acros Organics); 10 mM Tris (Sigma-Aldrich) solution with pH 7.2) at 37 °C for 3 min. After a final wash with PBS + 2% FCS, cells were resuspended in PBS + 2% FCS and live cells were counted in a Bürker chamber with trypan blue (VWR).

### Flow cytometry analysis

Three million cells per sample were washed with PBS and incubated for 15 min at RT with Zombie Aqua (BioLegend) or Zombie UV (BioLegend), together with Mouse Fc block (#130092575, Miltenyi Biotec). After washing with PBS + 2% FCS + 2 mM EDTA, cells were stained with fluorescently labeled monoclonal antibodies (Appendix Table S2) resuspended in PBS with 1:10 Brilliant stain buffer (BD Biosciences) for 20 min at 4 °C. Cells were washed with PBS and analyzed with a BD LSR Fortessa Flow cytometer (BD Biosciences). Results were analyzed with FlowJo v10 software (FlowJo LLC, BD). Cells were gated according to the strategies in Appendix Figs. S12 and 13.

### Histological analysis

After fixation, liver tissues were dehydrated by applying increasing ethanol concentrations in the Excelsior MS tissue processor (Thermo Fisher Scientific). Thereafter, tissues were embedded in paraffin with the HistoStar Workstation (Thermo Fisher Scientific) and 5 µm-thick tissue sections were made using with Microm HM 355S microtome (Thermo Fisher Scientific). Tissue sections were stained with hematoxylin and eosin (Abcam) and examined with a Leica DM 2000 light microscope (Leica Microsystems). Pictures were taken with magnification ×40 and captured using the LAS V4.2 software (Leica Microsystems). A blinded scoring system was implemented to quantify liver histopathology. The scoring system

was modified from Knackstedt et al and can be found in Appendix Table S3 (Knackstedt et al, 2019). Each parameter received an individual score between zero and three. The total histopathological score for each animal is the sum of all individual scores.

### RNA isolation and RT-qPCR

Tissues were mechanically homogenized in RLT buffer with the use of the Precellys 24 tissue homogenizer (Bertin Technologies). Next, RNA was extracted with the QIAGEN RNeasy Mini Kit (QIAGEN) according to the manufacturer's protocol. RNA concentration and purity were evaluated with the CLARIOSTAR plus plate reader (BMG Labtech). cDNA was then synthesized from the extracted RNA by using the High-Capacity cDNA Reverse Transcription kit (Applied Biosystems). To detect targeted gene expression levels, TaqMan Fast Universal PCR master mix (Applied Biosystems) was used in combination with specific primers (Integrated DNA Technologies; Appendix Table S4). The relative mRNA expression was determined as $2^{-\Delta\Delta CT}$, i.e., normalized to the mean $2^{-CT}$ value of the uninfected control WT mice and to the $2^{-CT}$ value of the 18S housekeeping gene.

### Liver RNA sequencing and data analysis

Liver RNA Sequencing (RNAseq) was performed by the Genomics Core UZ Leuven. RNA samples obtained from the liver of three to six individual mice per experimental condition were used. Libraries were prepared with the QuantSeq 3′ mRNA-Seq Library Prep Kit (LexoGen). Single-end sequencing was performed by the HiSeq 4000 sequencer (Illumina). 2 million 50 base pair reads per sample were sequenced. Quality control of raw reads was performed with FastQC v0.11.7. Adapters were filtered with ea-utils fastq-mcf v1.05. Splice-aware alignment was performed with HISAT2 against the reference genome mm10 using the default parameters. Reads mapping to multiple loci in the reference genome were discarded. Resulting BAM alignment files were handled with Samtools v1.5. Quantification of reads per gene was performed with HT-seq Count v0.10.0, Python v2.7.14. Count-based differential expression analysis was done with R-based (The R Foundation for Statistical Computing, Vienna, Austria) Bioconductor package DESeq2. Reported $P$ values were adjusted for multiple testing with the Benjamini–Hochberg procedure, which controls false discovery rate (FDR). Genes with a log2 fold change of at least |1| and an adjusted $P$ value of <0.05 were considered differentially expressed and selected for further analysis. Normalized counts and DESeq2 results can be found in Dataset EV1. Transcription factor enrichment analysis was performed by using the online Chea3 tool (https://maayanlab.cloud/chea3/) with the ENCODE, ReMap and literature-curated databases.

### Whole-blood transcriptome analysis: publicly available dataset

Publicly available RNA-sequencing data (accession number E-MTAB-6413) were obtained from the ArrayExpress database (EMBL-EBI). This dataset, originally published by (Lee et al, 2018), contains whole-blood transcriptomic profiles from pediatric malaria patients diagnosed with either cerebral malaria and hyperlactatemia (>5 mM; referred to as CH) or uncomplicated malaria (UM). Differential gene expression analysis was performed to compare CH and UM patient groups. The resulting differentially expressed genes were used for downstream enrichment analyses. Pathway enrichment analysis was conducted using

the WikiPathways database through the ShinyGO 0.80 platform. Transcription factor enrichment was performed using the literature-curated ChIP-seq database in the ChEA3 tool. In addition, signal transduction pathway analysis was carried out via the INOH (Integrating Network Objects with Hierarchies) database, also accessed through ShinyGO 0.80.

### Protein extraction and western blot

For protein extraction, tissues were lysed with modified radio-immunoprecipitation assay buffer (RIPA; (50 mM Tris, 150 mM NaCl, 1%v/v NP40, 0.25%v/v sodium deoxycholate, 1 mM EDTA, pH 7.4) containing protease (#11836170001, Roche) and phosphatase inhibitor cocktails (#P5726 and #P0044, Sigma-Aldrich). Then, samples were homogenized with the Precellys 24 tissue homogenizer (Bertin Technologies) and centrifuged (13,000 × g, 30 min, at 4 °C). Total protein concentrations were determined using the Pierce™ BCA Protein Assay Kit (Thermofisher) according to the manufacturer's instructions. Next, protein samples (25–50 μg per sample) were resolved on SDS-PAGE (4-12% or 10% Tris-Glycin Novex™ WedgeWell™; Thermofisher) and transferred onto nitrocellulose membranes using a wet transfer system. Membranes were blocked in 5% bovine serum albumin (BSA; Albumin Fraktion V, Carl Roth) or 5% non-fat dry milk (NFDM; BioRad) diluted in Tris-Buffered Saline Tween (TBST) buffer for 1 h at RT. Membranes were incubated with primary antibodies (Appendix Table S5) overnight at 4 °C. After washing with TBST buffer, membranes were incubated with donkey anti-rabbit horseradish peroxidase (HRP)-linked secondary antibody (#AB10015282, Jackson ImmunoResearch). To visualize protein bands, membranes were exposed to a chemiluminescent substrate (SuperSignal West Pico PLUS chemiluminescent substrate or SuperSignal West Femto Maximum Sensitivity Substrate; Thermo Fisher Scientific). Protein bands were digitally captured by the Fusion Solo S (Vilber Lourmat). Next, membranes were stripped with stripping buffer (0.1 M glycine; pH 2.8) and stained for total protein content with InstantBlue® Coomassie Protein Stain (Abcam). For semiquantitative analysis, immunodetected protein band intensities were measured using ImageJ software and normalized towards housekeeping protein expression or total protein content, as indicated.

## Statistical analysis

The data were analyzed using GraphPad PRISM software (version 9.5.1). To determine statistical differences between groups, the nonparametric two-tailed Mann–Whitney $U$ test was performed, followed by the Holm–Bonferroni correction for multiple testing. Significance levels for the survival curves were calculated with the Log-rank (Mantel–Cox) test. $P$ values were indicated as follows: $*P < 0.05$, $**P < 0.01$, and $***P < 0.001$. For bar graphs, the bar indicates the median, and each datapoint represents an individual mouse. For line graphs, the mean and standard error of the mean (SEM) are shown, or the longitudinal measurements of an individual mouse. Statistical differences compared to the corresponding uninfected control group are indicated with an asterisk above the individual data sets and horizontal lines with an asterisk on top indicate significant differences between groups. For correlation analysis, the nonparametric two-tailed Spearman correlation was calculated.

**The paper explained**

**Problem**

Severe hypoglycemia in complicated malaria is closely linked to lethality, even with optimal care. The underlying cause of this critical drop in blood glucose remains unclear, and there are currently no available adjunctive treatments for complicated malaria.

**Results**

Our study shows that the glucocorticoid pathway is critical to avoid the malaria-induced hypoglycemia. When the glucocorticoid pathway is deficient, overactivation of the JAK/STAT pathway leads to lethal hypoglycemia and hyperinflammation in malaria-infected mice. Treatment with the JAK/STAT inhibitor ruxolitinib successfully prevented hypoglycemia and hyperinflammation and improved survival without affecting parasitemia.

**Impact**

We identify JAK/STAT overactivation as a key driver of fatal hypoglycemia and hyperinflammation in malaria. Targeting this pathway with ruxolitinib may offer a novel therapeutic approach to prevent severe metabolic complications in malaria, potentially saving lives.

## Data availability

The RNAseq dataset produced in this study has been deposited in the NCBI Gene Expression Omnibus database under accession number: GSE268268.

The source data of this paper are collected in the following database record: biostudies:S-SCDT-10_1038-S44321-025-00264-w.

## Peer review information

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

## Acknowledgements

The authors would like to thank the Genomic core of the KU Leuven for performing the RNA sequencing, Willy Gsell for his contribution in optimizing the PET/MRI protocol for our study, Ghislain Opdenakker for the support, critical reading and useful suggestions and Kirsten Proost for assisting with performing western blots. This study received funding from the Research Foundation Flanders (F.W.O.-Vlaanderen, project G0C9720N and G066723N), the Research Fund of KU Leuven (project C14/23/143). Leen Vandermosten obtained a junior postdoctoral fellowship from F.W.O.-Vlaanderen. Hendrik Possemiers obtained an aspirant PhD fellowship from F.W.O.-Vlaanderen. Emilie Pollenus obtained a L'Oréal-Unesco Women for Sciences PhD fellowship. Jan Tuckermann received support from the Deutsche Forschungsgemeinschaft for the CRC1506 (Aging at Interfaces) Project number 450627322, TRR369 (Dione) Projektnummer 501752319. Sabine Vettorazzi received support from the Deutsche Forschungsgemeinschaft for the research grant project number 497680553 and 537908488.

## Author contributions

**Fran Prenen**: Conceptualization; Data curation; Software; Formal analysis; Validation; Investigation; Visualization; Methodology; Writing—original draft; Project administration; Writing—review and editing. **Leen Vandermosten**: Conceptualization; Data curation; Software; Formal analysis; Validation; Investigation; Visualization; Methodology; Writing—original draft; Project administration; Writing—review and editing. **Sofie Knoops**: Investigation; Project administration; Writing—review and editing. **Emilie Pollenus**: Investigation; Writing—review and editing. **Hendrik Possemiers**: Investigation; Writing—review and editing. **Pauline Dagneau de Richecour**: Investigation; Writing—review and editing. **Giorgio Caratti**: Investigation; Methodology; Writing—review and editing. **Christopher Cawthorne**: Investigation; Methodology; Writing—review and editing. **Sabine Vettorazzi**: Investigation; Methodology; Project administration; Writing—review and editing. **Yevva Cranshoff**: Investigation; Writing—review and editing. **Dominique Schols**: Resources; Writing—review and editing. **Sandra Claes**: Investigation; Writing—review and editing. **Christophe M Deroose**: Methodology; Writing—review and editing. **Uwe Himmelreich**: Methodology; Writing—review and editing. **Jan Tuckermann**: Resources; Funding acquisition; Methodology; Writing—review and editing. **Philippe E Van den Steen**: Conceptualization; Resources; Supervision; Funding acquisition; Methodology; Writing—original draft; Project administration; Writing—review and editing.

Source data underlying figure panels in this paper may have individual authorship assigned. Where available, figure panel/source data authorship is listed in the following database record: biostudies:S-SCDT-10_1038-S44321-025-00264-w.

## Disclosure and competing interests statement

The authors declare no competing interests.

# Expanded View Figures

**Figure EV1.   Validation of tamoxifen treatment for the induction of global GR deletion.**

Tamoxifen-inducible WT and GRiKO mice were treated with vehicle or 10 mg/kg tamoxifen for 5 consecutive days. Organs were collected at 1 week, 2 weeks or 4 weeks after the last tamoxifen treatment was given. Protein expression of GRα were determined in (**A**) spleen, (**B**) liver, (**C**) lung, (**D**) whole brain, (**E**) adrenal gland, (**F**) pancreas, (**G**) quadriceps muscle, (**H**) white adipose tissue (WAT) and (**I**) kidney extracts. For (**A–D**): percentage of reduction in GRα protein expression in tamoxifen-treated GRiKO group relative to vehicle-treated GRiKO group. For (**E–I**): percentage of reduction in GRα protein expression in tamoxifen-treated GRiKO group relative to tamoxifen-treated WT group is shown. Representative blots are shown together with the semiquantitative determination of band intensities. Each datapoint represents an individual mouse. Data from one or two independent experiments. Bar graph represent median per group. Statistical differences were calculated with the Mann–Whitney *U* test. ns: $P > 0.05$. *$P < 0.05$. Source data are available online for this figure.

▶

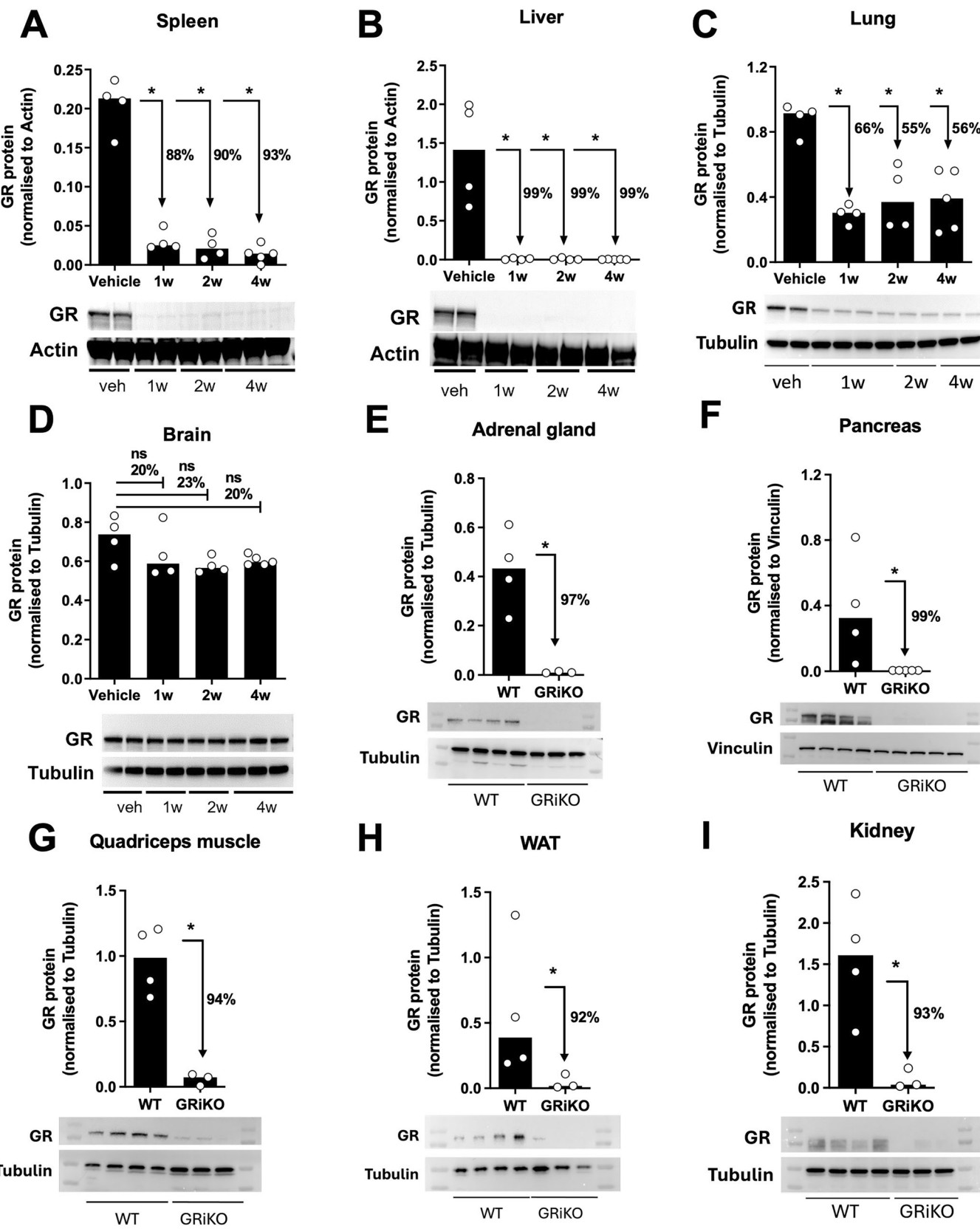

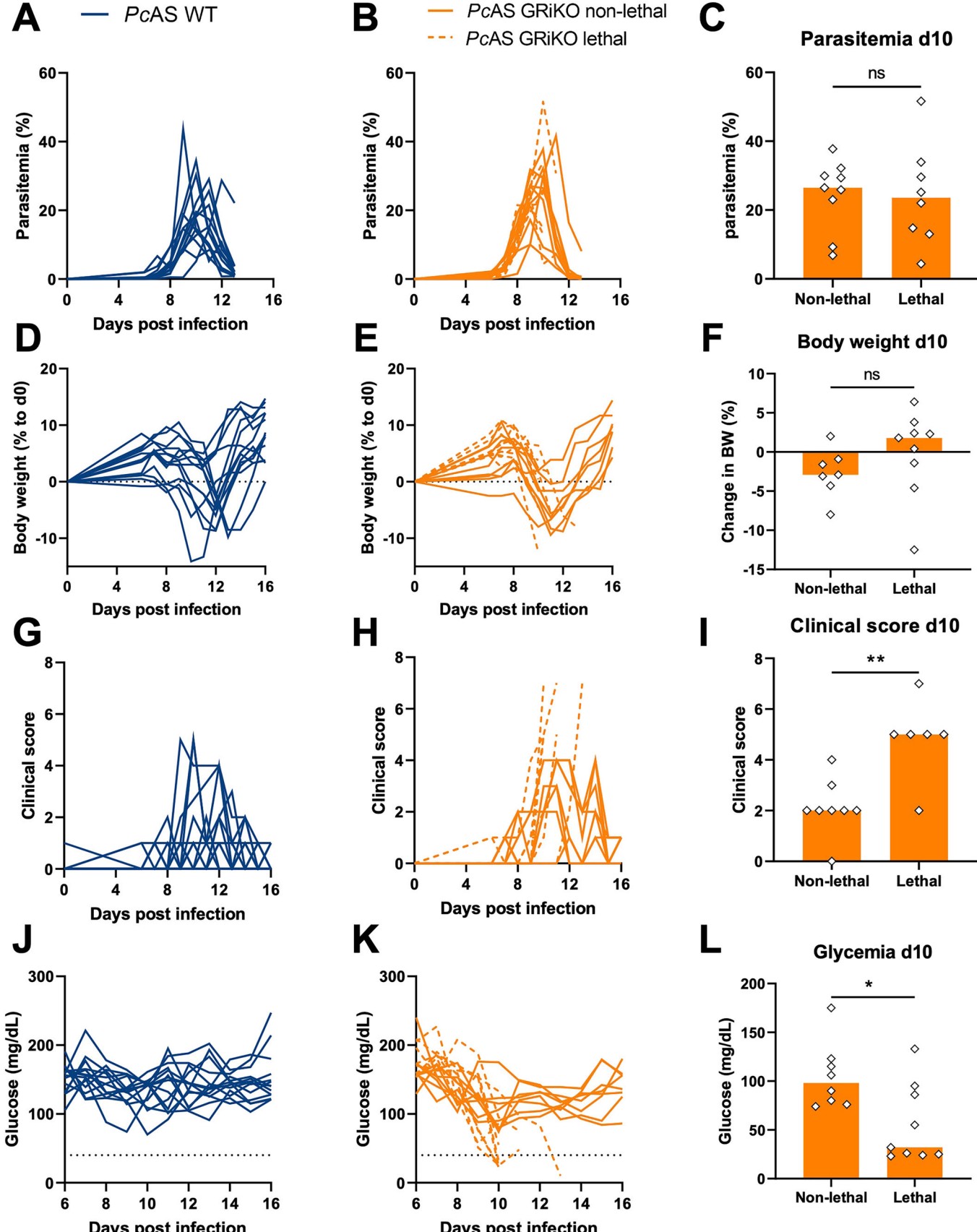

◄ **Figure EV2.  The observed lethality in infected GRiKO mice is related to clinical score and glycemia, and not to parasitemia or body weight.**

WT and GRiKO mice were infected with *Pc*AS parasites. (**A–C**) Parasitemia levels, (**D–F**) body weight, (**G–I**) clinical score and (**J–L**) glycemia were determined each day, from 6 dpi onwards. For each parameter, longitudinal data is represented. A statistical comparison was made for all four parameters (measured at 10 dpi) between lethal and non-lethal infected GRiKO mice (data from the same experiments as in Fig. 1C-E, I-J and Appendix Fig. S1C-E). Statistical differences were calculated with the Mann–Whitney *U* test. Each datapoint or line represents an individual mouse. Solid lines: non-lethal mice, and striped lines: lethal mice. Data from three independent experiments. Bar graph represent median per group. ns: $P > 0.05$, *$P < 0.05$, **$P < 0.01$.

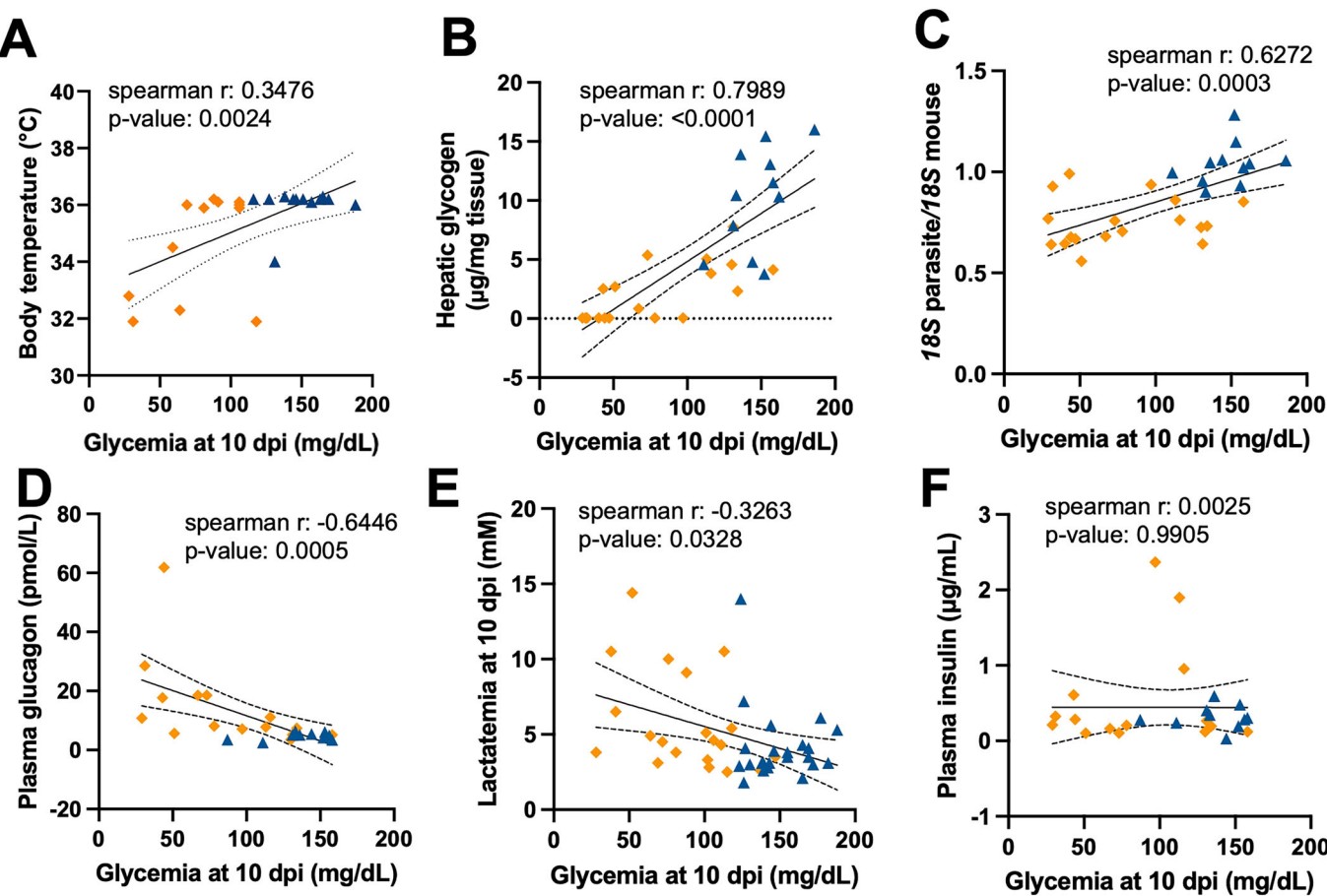

**Figure EV3. Body temperature, hepatic glycogen, plasma glucagon, lactatemia and hepatic parasite load correlate with glycemia levels.**

WT and GRiKO mice were infected with *Pc*AS parasites. At 10 dpi, glycemia, body temperature, hepatic glycogen, plasma glucagon levels and lactatemia were determined. Correlations were assessed between glycemia and (A) body temperature, (B) hepatic glycogen, (C) hepatic parasite load, (D) plasma glucagon, (E) lactatemia and (F) plasma insulin with a nonparametric Spearman test. ▲: infected WT mice and ◆: infected GRiKO mice. Each symbol represents an individual mouse. Data from two individual experiments.

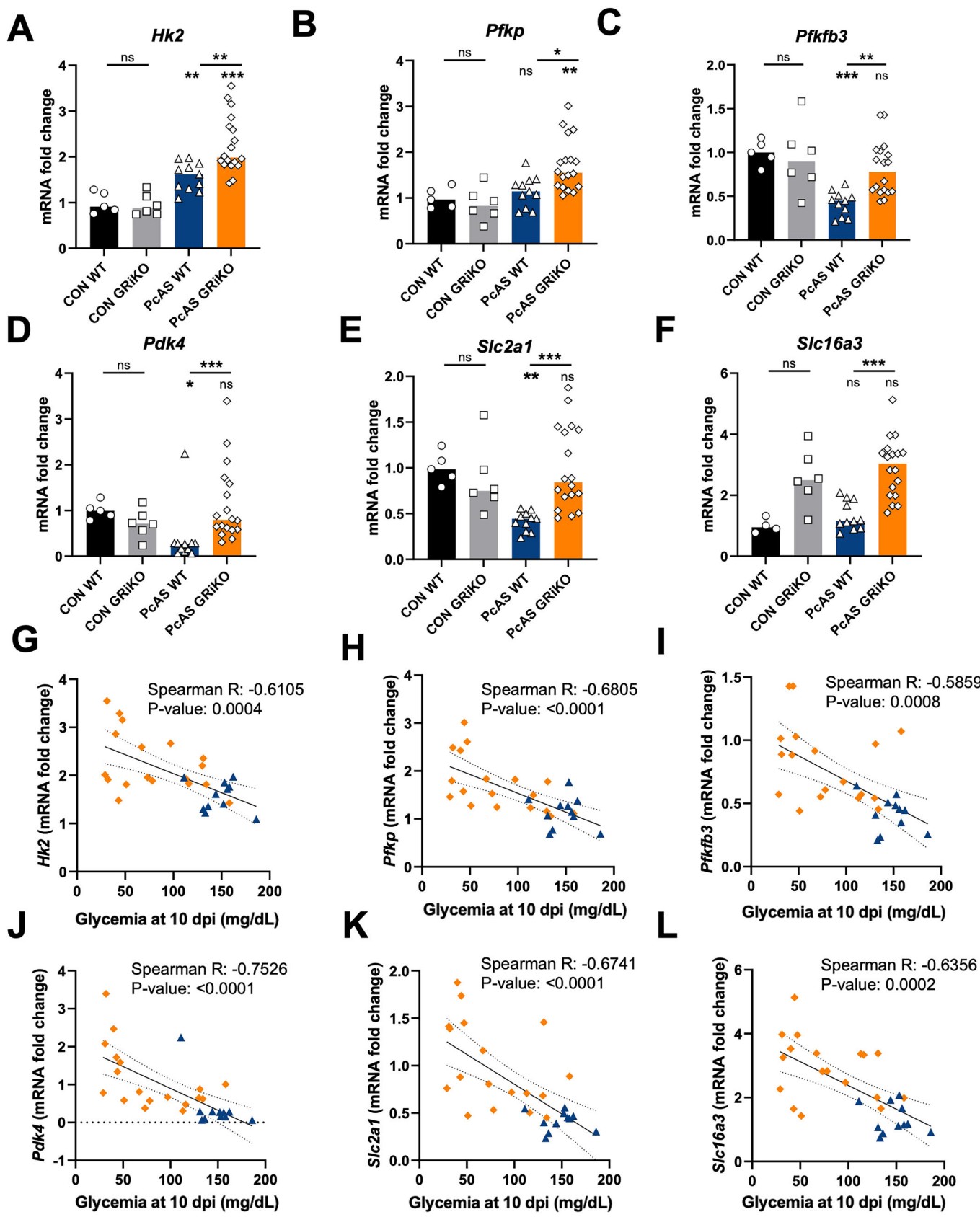

◀ **Figure EV4.  GR signaling inhibits glycolytic gene expression in spleen during infection.**

WT and GRiKO mice were infected with *Pc*AS parasites and spleens were collected at 10 dpi. qPCR was performed to determine mRNA expression of (**A**) *Hk2*, (**B**) *Pfkp*, (**C**) *Pfkfb3*, (**D**) *Pdk4*, (**E**) *Slc2a1* and (**F**) *Slc16a3*. Data were normalized towards mouse-specific *18S* and the average expression of CON WT mice. Correlation analyses were performed between glycemia and (**G**) *Hk2*, (**H**) *Pfkp*, (**I**) *Pfkfb3*, (**J**) *Pdk4*, (**K**) *Slc2a1* and (**L**) *Slc16a3* with the nonparametric Spearman test. Statistical differences were calculated with the Mann–Whitney *U* test with Holm–Bonferroni correction. ns: $P > 0.05$, $*P < 0.05$, $**P < 0.01$, $***P < 0.001$. Asterisks above data points indicate significant differences between infected and non-infected mice, asterisks above a horizontal line show significant differences between genotype. Each symbol represents an individual mouse and bar graph represent median per group. ▲: infected WT mice and ◆: infected GRiKO mice. Data from two independent experiments.

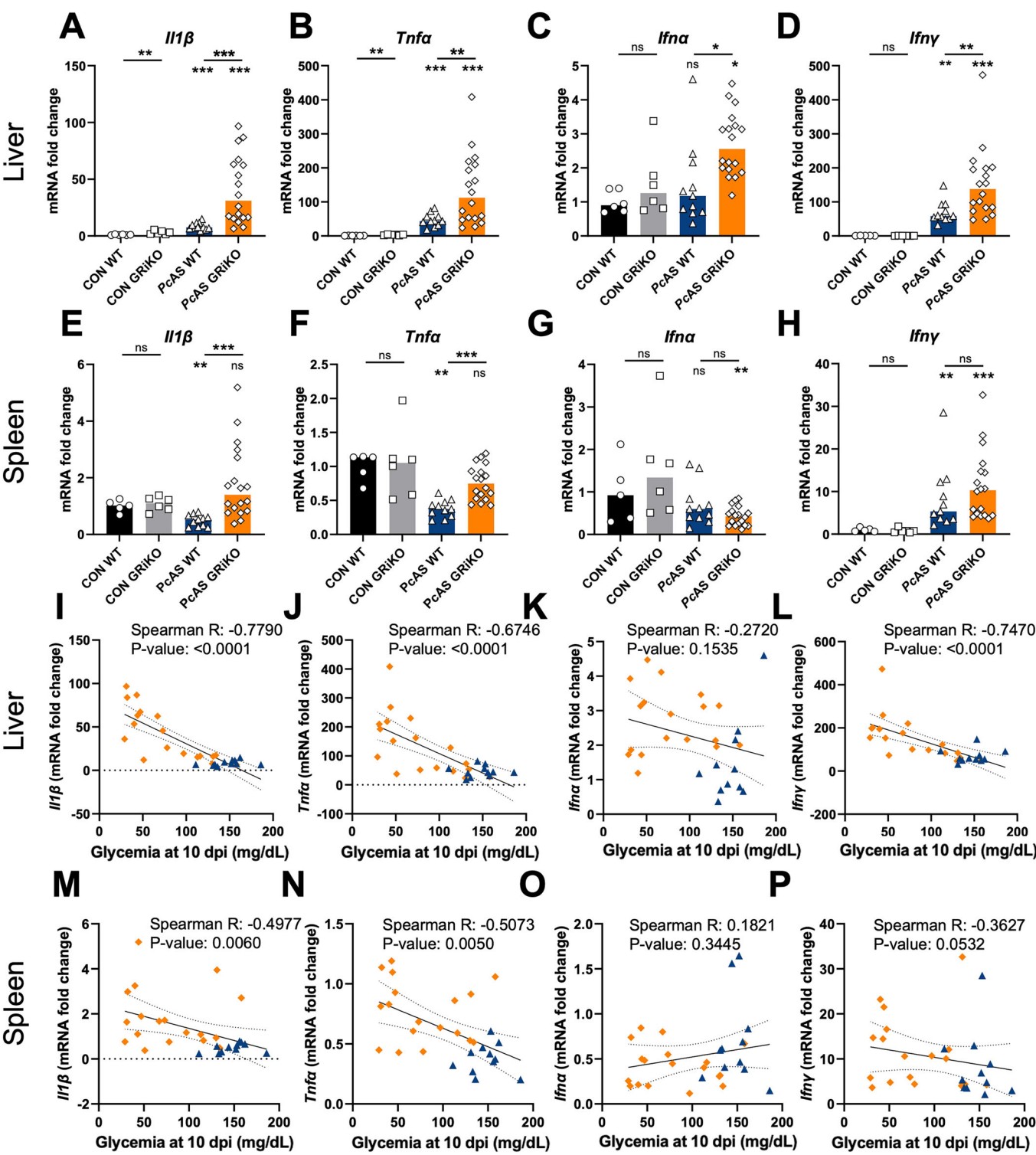

**Figure EV5. GR signaling prevents cytokine storm in liver and spleen of *Pc*AS-infected mice.**

WT and GRiKO mice were infected with *Pc*AS parasites. At 10 dpi, livers and spleens were collected and expression levels of (**A, E**) *Il1β* (**B, F**) *Tnfα*, (**C, G**) *Ifnα and* (**D, H**) *Ifnγ* were measured with qPCR. Data were normalized towards mouse-specific *18S* and the average expression of CON WT mice. (**I–P**) Correlations between glycemia levels and the cytokine expressions were assessed with the nonparametric Spearman test. Each symbol represents an individual mouse and bar graph represent median per group. ▲: infected WT mice and ◆: infected GRiKO mice. Statistical differences were calculated with the Mann–Whitney *U* test with Holm–Bonferroni correction. ns: *P* > 0.05, *P < 0.05, **P < 0.01, ***P < 0.001. Asterisks above data points indicate significant differences between infected and non-infected mice, asterisks above a horizontal line show significant differences between genotype. Data from two independent experiments.

