## [Peer Review File · EMBO Molecular Medicine]

Ruxolitinib protects glucocorticoid receptor knockout mice from lethal malaria-induced hypoglycemia

Fran Prenen, Vandermosten Leen, Sofie Knoops, Emilie Pollenus, Hendrik Possemiers, Pauline Dagneau de Richecour, Giorgio Caratti, Christopher Cawthorne, Sabine Vettorazzi, Yevva Cranshoff, Dominique Schols, Sandra Claes, Christophe Deroose, Uwe Himmelreich, Jan Tuckermann, and Philippe Van den Steen

Corresponding author(s): Philippe Van den Steen (philippe.vandensteen@kuleuven.be)

Review Timeline:

Submission Date:	4th Oct 24
Editorial Decision:	19th Nov 24
Revision Received:	20th May 25
Editorial Decision:	3rd Jun 25
Revision Received:	11th Jun 25
Accepted:	12th Jun 25

Editor: Jingyi Hou

Transaction Report:

19th Nov 2024

Dear Prof. Van den Steen,

Thank you again for submitting your work to EMBO Molecular Medicine. First of all, I would like to apologize for the somewhat slow process, which was due to the late arrival of referees' reports. We have now heard back from the three referees who evaluated your manuscript. As you will see from the reports below, the referees find the topic of your study of interest. However, they raise a series of concerns, which should be convincingly addressed in a major revision of the present manuscript.

The referees' recommendations are relatively straightforward, so there is no need to reiterate the points listed below. During our pre-decision cross-commenting (in which the referees are given a chance to make additional comments, including on each other's reports), Referees #2 added, "I agree with my colleague's reviews. Concerning the other models to be used, I think they are necessary, as Referee#1 states the lethality observed can be due to cytokine storm/hyperinflammation. Other models known to be less acute could answer this. Nevertheless, authors can answer without difficulty; the proposed models are already used by the group." In light of these comments, we would ask you to include at least some experimental data from an additional animal model to strengthen the conclusions of the study. All other issues need to be carefully addressed as well.

We would welcome the submission of a revised version within three months for further consideration. Please feel free to contact me in case you would like to discuss in further detail any of the issues raised by the referees.

As you may already know, our editorial policy allows in principle a single round of major revision, and it is therefore essential to provide responses to the reviewers' comments that are as complete as possible.

I look forward to receiving your revised manuscript.

Kind regards,
Jingyi

Jingyi Hou
Editor
EMBO Molecular Medicine

We require:

- 1) A .docx formatted version of the manuscript text (including legends for main figures, EV figures and tables). Please make sure that the changes are highlighted to be clearly visible.
- 2) Individual production quality figure files as .eps, .tif, .jpg (one file per figure). For guidance, download the 'Figure Guide PDF': (<https://www.embopress.org/page/journal/17574684/authorguide#figureformat>).
- 3) A .docx formatted letter INCLUDING the reviewers' reports and your detailed point-by-point responses to their comments. As

part of the EMBO Press transparent editorial process, the point-by-point response is part of the Review Process File (RPF), which will be published alongside your paper.

4) A complete author checklist, which you can download from our author guidelines (<https://www.embopress.org/page/journal/17574684/authorguide#submissionofrevisions>). Please insert information in the checklist that is also reflected in the manuscript. The completed author checklist will also be part of the RPF.

6) It is mandatory to include a 'Data Availability' section after the Materials and Methods. Before submitting your revision, primary datasets produced in this study need to be deposited in an appropriate public database, and the accession numbers and database listed under 'Data Availability'. Please remember to provide a reviewer password if the datasets are not yet public (see <https://www.embopress.org/page/journal/17574684/authorguide#dataavailability>).

.

- the medical issue you are addressing,

- the results obtained and

- their clinical impact.

This may be edited to ensure that readers understand the significance and context of the research. Please refer to any of our

published articles for an example.

12) Author contributions: You will be asked to provide CRediT (Contributor Role Taxonomy) terms in the submission system. These replace a narrative author contribution section in the manuscript.

13) A Conflict of Interest statement should be provided in the main text.

14) Every published paper now includes a 'Synopsis' to further enhance discoverability. Synopses are displayed on the journal webpage and are freely accessible to all readers. They include a short stand first (maximum of 300 characters, including space) as well as 2-5 one-sentence bullet points that summarize the paper. Please write the bullet points to summarize the key NEW findings. They should be designed to be complementary to the abstract - i.e. not repeat the same text. We encourage inclusion of key acronyms and quantitative information (maximum of 30 words / bullet point). Please use the passive voice. Please attach these in a separate file or send them by email, we will incorporate them accordingly.

15) All Materials and Methods need to be described in the main text using our 'Structured Methods' format. According to this format, the Methods section includes a Reagents and Tools Table (listing key reagents, experimental models, software and relevant equipment and including their sources and relevant identifiers) followed by a Methods and Protocols section describing the methods, ideally using a step-by-step protocol format. The aim is to facilitate adoption of the methodologies across labs.

Please download and fill our Reagents and Tools Table template (.docx), which you can find in our author guidelines: <https://www.embopress.org/page/journal/17574684/authorguide#structuredmethods>

***** Reviewer's comments *****

Referee #1 (Remarks for Author):

The paper explores the mechanism of disease tolerance in malaria, revealing a striking role of glucocorticoid signaling in regulation of hypoglycemia. This effect is shown to be mediated via JAK/STAT signaling and an inhibitor is identified as a potential candidate for adjunctive therapy.

The model is appropriate, and experiments are well designed and skillfully executed. Statistical analysis is robust and conclusions are sound. The paper is well written. This study is valuable and should be published.

Comments/questions:

The paper would benefit from demonstrating human relevance. Since the authors did carry out a related human study, can they include relevant data to support the described mechanism? For example, STAT phosphorylation in leukocytes and how this correlates with hypoglycemia?

Do you have data on depletion levels for the cell-specific knockdowns? Could partial depletion be the reason for no cell-specific effect on hypoglycemia?

Are mice dying of hyperlactatemia? Or are the observed lactate values in infected GRiko mice within normal range?

Similarly - are mice dying of cytokine storm? How do you distinguish between hypoglycemia and hyperinflammation as cause of increased mortality?

Limited clinical trials with dexamethasone failed to provide protection for severe malaria - please comment in discussion

Referee #2 (Comments on Novelty/Model System for Author):

The model used is a model of infection with *P. chabaudi* AS, which, as the authors mentioned, typically induces only mild symptoms despite high parasitemia. It would be interesting to see what happens in another model, for example, *P. yoelii* NL (nonlethal) or a lethal model used by the authors in their cited papers, such as *P. berghei* NK65. This would increase the impact of their conclusions.

Referee #2 (Remarks for Author):

In their research study, "Ruxolitinib protects glucocorticoid receptor knockout mice from lethal malaria-induced hypoglycemia," Prenen et al. demonstrated the role of the Glucocorticoid receptor in tolerance during *P. chabaudi* AS infection. The study is clearly written and has clear objectives, and the results support the conclusions in most cases. Ruxolitinib treatment adds a potential translational interest. Nevertheless, I have some major comments that should be addressed before publication:
Fig. 1.

Expression of GCr on KO mice: The authors check the expression of GCr after tamoxifen administration in the spleen and liver. They say that there was a global deletion of the GCr. Nevertheless, they don't check the expression in the adrenal gland and other organs, such as the lungs.

GRiko mice characterization: SNP was only performed in six mice. Why were they not performed in all the mice used? Can the authors clarify this sentence?

Parasitemia in WT vs. KO mice: the authors only measured the parasitemia in the peripheral blood. What is the parasite load on the tissues? A BLI follow-up of the infection is necessary. What does happen with KO mice that don't succumb to the infection? Can the authors show separate graphs for the ko mice that survive? Parasitemia, clinical score, glycemia, liver and spleen weight... Are they not expressing the GR?

-Fig 2. Glucose uptake measurement by NMR. Was the measure only made in a few animals? What did happen with the GRiko mice that did not succumb? Did they have the same accumulation in the liver and spleen?

-Fig 3. The authors need to show the expression of GR in the conditional ko mice in the different immune cells to probe the efficacy of the cre.

-Fig 5.

It is not clear if the ChEA3 analysis was performed with the model's use or if it was performed using databases. Can the authors clarify this? Did they perform the ChIPseq?

Fig 5F: is the correlation calculated with the pSTAT/pSTAT3 data from the WB on E? WB is a semiquantitative analysis. The authors should use a quantitative method to measure the pSTAT/pSTAT3 and make the correlation.

Fig 6: The absence of phenotype in conditional mice don't match the results in figure 6 where inflammatory cytokines are upregulated. Experiments with WT or KO mice receiving KO or WT BM cells would reinforce the author's conclusions.

Fig 7: Although the use of Ruxolitinib is innovative and adds a translational aspect to the study, I have several comments in this regard:

-Survival of Ruxolitinib-treated mice compared to GRi ko that don't die

-The glucose uptake experiments, as in Fig 2, should be done

-Immune cells and cytokine expression on the treated mice?

-What is the impact of the Ruxolitinib treatment on other murine malaria models? *Pyoelii* NL or *Pberghei* NK65?

In the T-cell response experiments, the Effector phenotype is assessed, but are these antigen-responding cells?

Referee #3 (Comments on Novelty/Model System for Author):

Prenen & Vandermosten et al. gathered an impressive amount of work investigating the role of glucocorticoid receptor in disease tolerance during *Plasmodium chabaudi* infection.

Their data show that in the absence of GR induced by tamoxifen in GRiKO infected mice more often develop lethal hypoglycemia, and that their liver and spleen retain glucose. They tested single KO lacking GR in hepatocyte, T-cells, or myeloid cells specifically and observed no increased lethality, but combinations of multiple KO were not tested.

These authors then investigated liver and spleen gene expression in on infected and control mice with and without tamoxifen treatment to abolish GR, and detected the DEGs were associated with JAK/STAT activation and show an increased ratio of phosphorylated STAT3 over total, supporting increased STAT3 activation in GR deficient mice.

Then they used Ruxolitinib to inhibit stat3 in infected mice with and without GR and observe a recovery of the lethal impact of Pcc in mice.

I believe their efforts are worthy of publication and I have only a few suggestions that I think may help the reader follow and improve possible discussion of future work

Ln 56. what tissues were used to detect the elevated GCs in malaria mice and humans? Be clear

Ln. 83 I'd mention briefly in the result section how does tamoxifen-inducible glucocorticoid receptor knockout mice work, and I

suggest adding Ref in the result and not only in the methods
GRiKO mice is maybe not the best name

As a reader I prefer to have figures describe in independent subheadings, but the editors will know better what the publication standard is.

Fig. S3c parasitaemia could be compared at day of death vs that day of surviving mice, instead of d10

Is Fig 2b & C done from calculations images in Fig 2 A or from the organs removed? Should be clear from reading the result section too.

Fig. S5 was done at d10 in sacrificed mice, but seems unclear what their survival outcome would be, was the clinical score recorded? Representative images are shown but how many mice were used, and were there 2 different outcomes 50/50 as would be expected from the clinical score and lethality description earlier.

Ln 175 & 180 I'd mention that the only single KO and no multiple combinations were not tested

Ln 253 I'd mention that ruxolitinib has been shown to inhibit stat3 also, and not only that it is a selective JAK1/2 inhibitor, and I'd show the clinical score on the main Fig7 and not on the Fig. S

I'd mention the symptoms that were present when the treatment started, and would discuss how late could the treatment be applied, thinking of future clinical application for children with severe malaria.

Point-by-point reply**Referee #1:**

The paper explores the mechanism of disease tolerance in malaria, revealing a striking role of glucocorticoid signaling in regulation of hypoglycemia. This effect is shown to be mediated via JAK/STAT signaling and an inhibitor is identified as a potential candidate for adjunctive therapy. The model is appropriate, and experiments are well designed and skillfully executed. Statistical analysis is robust and conclusions are sound. The paper is well written. This study is valuable and should be published.

Comments/questions:

The paper would benefit from demonstrating human relevance. Since the authors did carry out a related human study, can they include relevant data to support the described mechanism? For example, STAT phosphorylation in leukocytes and how this correlates with hypoglycemia?

We appreciate the referee's insightful suggestion regarding demonstrating human relevance. While we currently lack access to samples to directly correlate STAT phosphorylation in leukocytes from severe malaria patients with hypoglycemia, we explored the literature to strengthen the translational relevance of our findings. Notably, Lee et al. (2018, *Science Translational Medicine*) performed transcriptomic analysis of whole blood from cerebral malaria patients with hyperlactatemia compared to uncomplicated malaria patients. Through transcription factor enrichment analysis (TFEA, with ChEA3 tool) of differentially expressed genes discriminating these groups, we observed STAT3 among the top two enriched transcription factors. Supporting this, signal transduction pathway analysis with the INOH database also revealed enrichment in JAK1 and JAK2 among the top 10 enriched pathways. Furthermore, pathway analysis using WikiPathways identified glycolysis as the top enriched pathway. These data strongly support our hypothesis that the JAK/STAT3 pathway plays a critical role in malaria pathophysiology and that targeting this pathway could have translatable therapeutic potential, particularly for severe cases characterized by hypermetabolic states. We have created an additional main figure (fig. 7) and a separate paragraph for these data in the result section.

Do you have data on depletion levels for the cell-specific knockdowns? Could partial depletion be the reason for no cell-specific effect on hypoglycemia?

The efficiency of the conditional cell-specific GR KO mice have been characterized previously in detail, in exactly the same mouse strains as used here.

GR^{Lck-cre} mice: Baschant et al. showed that in GR^{Lck-cre}, deletion efficiency was ~99% in sorted T cells (both CD4 and CD8), whereas poor or no recombination was observed in myeloid cells and B cells (Baschant et al., PNAS 2011, 108(48):19317-22).

GR^{Lck-cre} mice: Tuckermann et al. assessed the recombination efficiency in GR^{LysM-Cre} mice by crossing GRLysMCre mice with tk-loxP-enhanced GFP (tk-loxP-EGFP) reporter (RA/EG) animals (Tuckermann et al., 2007, JCI 117(5):1381-90). Cre-mediated recombination was determined by flow cytometry measurement of EGFP in cells isolated from bone marrow, spleen and lymph

nodes. Almost complete recombination of the loxP sites was achieved in neutrophils (more than 90%) and macrophages (70%), whereas recombination in LCs (43%), dendritic cells (16%), and mast cells (26%) was less effective. T and B cells were largely unaffected. Effective disruption of the GR gene in neutrophils was further demonstrated by the complete conversion of the GR^{flox} allele into the GR^{null} allele in FACS-sorted EGFP+GR-1+ Granulocytes showing a 92% and 70% efficiency in neutrophils and macrophages.

GR^{Alb-Cre} mice: Southern blot indicated highly efficient recombination efficiency only in the liver, and immunohistochemical staining showed an almost complete loss of GR in hepatocytes (Kellendonk, 2000, *genesis* 26:151–153). Mueller et al. also showed efficient GR knockout by Western blotting in total liver extracts (Mueller et al., 2011, *Hepatology* 54(4):1398-409).

Overall, we think that the excellent recombination and knockout efficiency has been well documented, and we have added these relevant references in the text on lines 501-502. Therefore, although it is impossible to fully exclude that minor residual GR expression might have played a role, this seems highly unlikely.

Are mice dying of hyperlactatemia? Or are the observed lactate values in infected GRiKO mice within normal range?

Although a correlation between hyperlactatemia and hypoglycemia was noted, we do not think that hyperlactatemia is a main cause of death in these mice. First, while the observed hyperlactatemia is significant and pronounced, it remains below 15 mM and for most mice even below 10 mM. In contrast, *P. yoelii* XL infected mice develop hyperlactatemia levels of >17 mM (Georgiadou et al., 2022, *eLife*). Most importantly, ruxolitinib rescued mice from lethality and inhibited severe hypoglycemia, but had no effect on hyperlactatemia (Appendix fig. S10C). Combined, this clearly suggests that hyperlactatemia is not the main cause of death in these mice.

Similarly - are mice dying of cytokine storm? How do you distinguish between hypoglycemia and hyperinflammation as cause of increased mortality?

We thank the referee for this important and valid point. Indeed, our current data do not allow us to definitively distinguish whether hypoglycemia or hyperinflammation is the primary cause of death in GRiKO mice. Our initial interpretation focused on hypoglycemia, which is based on the known lethal effects of hypoglycemia. Plasma glucose levels below 50 mg/dL cause functional brain failure and coma, and levels below 20 mg/dL result in rapid brain death (Cryer et al., *J Clin Invest*, 2007). However, we have now also measured systemic cytokine levels and found them to be significantly elevated in infected GRiKO mice. In addition, we found that these levels are correlated strongly with glycemia, suggesting a link between metabolic and inflammatory dysregulation; and that ruxolitinib treatment was also capable of suppress IL-6 and IL-10 cytokine elevations upon infection (fig. 9).

Considering this, we have revised the manuscript to clarify that GRiKO mice likely succumb to a combination of severe hypoglycemia and systemic hyperinflammation, rather than hypoglycemia alone. We have updated the relevant sections in the title, abstract, introduction, results and discussion.

Limited clinical trials with dexamethasone failed to provide protection for severe malaria - please comment in discussion.

We have elaborated more on this in the discussion (line 429-433)

Referee #2:

The model used is a model of infection with *P. chabaudi* AS, which, as the authors mentioned, typically induces only mild symptoms despite high parasitemia. It would be interesting to see what happens in another model, for example, *P. yoelii* NL (nonlethal) or a lethal model used by the authors in their cited papers, such as *P. berghei* NK65. This would increase the impact of their conclusions.

Comments and questions:

In their research study, "Ruxolitinib protects glucocorticoid receptor knockout mice from lethal malaria-induced hypoglycemia," Prenen et al. demonstrated the role of the Glucocorticoid receptor in tolerance during *P.chaubaudi* AS infection. The study is clearly written and has clear objectives, and the results support the conclusions in most cases. Ruxolitinib treatment adds a potential translational interest. Nevertheless, I have some major comments that should be addressed before publication:

Fig.1. Expression of GCr on KO mice: The authors check the expression of GCr after tamoxifen administration in the spleen and liver. They say that there was a global deletion of the GCr. Nevertheless, they don't check the expression in the adrenal gland and other organs, such as the lungs.

You are correct that we did not initially check the expression of the glucocorticoid receptor (GR) in the adrenal gland. This is a valid point, as the adrenal gland regulates GC production. Therefore, we have now examined efficiency of GR deletion in the adrenal gland with western blot. In addition, we extended this analysis to other tissues including the kidney, pancreas, white adipose tissue, muscles and lung (liver, spleen and brain were already included in the initial manuscript). In most of the analyzed tissues, we detected efficient GR deletion, confirming the validity of our approach. In brains, almost no deletion occurred, which is in fact beneficial to avoid major effects on the hypothalamus – hypophysis – adrenal axis, as already previously stated in our paper. Furthermore, in the lungs, the deletion efficiency was around 60%, which is reasonable but not fully conclusive. Therefore, we cannot exclude that the residual GR expression in the lungs may have partially attenuated the phenotype of GR knockout in our study. We have added the relevant data to the manuscript in Fig. EV1 for clarity and completeness.

GRiko mice characterization: SNP was only performed in six mice. Why were they not performed in all the mice used? Can the authors clarify this sentence?

We appreciate the referee's comment. SNP analysis was performed on six mice which are representative for the whole breeding colony (including both WT and GRiKO mice). For each experiment, we used littermates originating from this colony. Since all the mice used were genetically similar, and because of the high cost per sample of this >2000 SNP analysis, we did not perform SNP analysis on the entire cohort.

Parasitemia in WT vs. KO mice: the authors only measured the parasitemia in the peripheral blood.

What is the parasite load on the tissues? A BLI follow-up of the infection is necessary. What does happen with KO mice that don't succumb to the infection?

We cannot perform bioluminescence imaging (BLI) due to ethical approval restrictions within the revision time. However, to address this concern, we calculated the parasite load in the circulation at 10 dpi by correcting parasitemia for the number of erythrocytes (Appendix S1A). Additionally, we measured the parasite 18S rRNA in the liver at 10 dpi using qPCR to assess the tissue-specific parasite burden (Appendix S1B and S10A). We added this data to the manuscript.

This methodology has an important advantage compared to BLI, as qPCR is highly quantitative and does not suffer from issues of black skin, which reduces the BLI detection according to the local thickness of the fur, and expression differences of the luciferase according to the parasite stage.

Can the authors show separate graphs for the ko mice that survive? Parasitemia, clinical score, glycemia, liver and spleen weight... Are they not expressing the GR?

We appreciate the referee's suggestion. To investigate the factors driving lethality, we have added Fig. EV2, where we compare parasitemia, clinical score, body weight, and glycemia at day 10 (d10) between surviving and non-surviving mice. We chose d10 because by d11, many mice had already died, and we no longer had data for them. In our analysis, we found significant differences only in clinical score and glycemia between the two groups. We did not observe significant differences in parasitemia or body weight that would distinguish surviving from non-surviving mice.

Regarding GR expression, the referee is indeed right that incomplete deletion of the GR might be suspected as the reason for the survival of some GRiKO mice. To assess whether GR was indeed not sufficiently deleted, we performed Western blot analysis for GR expression on liver extracts of surviving *PcAS*-infected GRiKO mice and compared it to surviving *PcAS*-infected GRiKO mice. However, our analysis shows complete absence of GR in surviving infected GRiKO mice, suggesting that survival is not determined by partial GR expression. We believe the outcome is likely due to a delicate balance of immune and metabolic factors influencing lethality. The results can be found in Appendix S2, and complementary text at line 134-137 in the manuscript.

-Fig 2. Glucose uptake measurement by NMR. Was the measure only made in a few animals? What did happen with the GRiKO mice that did not succumb? Did they have the same accumulation in the liver and spleen?

The glucose uptake measurements by PET/MRI were performed on 7 to 8 infected mice for both WT and GRiKO group (and 3 – 6 mice for the non-infected groups). All individual datapoints are visible in figure 2B and 2C. Since the infected mice exhibited varying glycemia levels at the time of imaging (ranging from 22-188 mg/dL), we additionally examined whether FDG uptake correlated with blood glucose levels. Interestingly, we observed a clear negative correlation between glycemia and FDG uptake in both the liver and spleen, suggesting that enhanced organ-specific glucose uptake may contribute to systemic hypoglycemia. The related data is provided in figure 2D-E of the revised manuscript.

We acknowledge that it would be valuable to assess whether increased FDG uptake is predictive of lethality. However, due to the use of a radioactive tracer, animals must be euthanized after imaging, in compliance with institutional radiation safety protocols. As a result, we were unable to perform longitudinal follow-up or survival analysis on these specific mice.

-Fig 3. The authors need to show the expression of GR in the conditional ko mice in the different immune cells to probe the efficacy of the cre.

The efficiency of the conditional cell-specific GR KOs have been characterized previously in detail, in exactly the same mouse strains as used here.

GR^{Lck-cre} mice: Baschant et al. showed that in GR^{Lck-cre}, deletion efficiency was ~99% in sorted T cells (both CD4 and CD8), whereas poor or no recombination was observed in myeloid cells and B cells (Baschant et al., PNAS 2011, 108(48):19317-22).

GR^{Lck-cre} mice: Tuckermann et al. assessed the recombination efficiency in GR^{LysM-Cre} mice by crossing GRLysMCre mice with tk-loxP-enhanced GFP (tk-loxP-EGFP) reporter (RA/EG) animals (Tuckermann et al., 2007, JCI 117(5):1381-90). Cre-mediated recombination was determined by flow cytometry measurement of EGFP in cells isolated from bone marrow, spleen and lymph nodes. Almost complete recombination of the loxP sites was achieved in neutrophils (more than 90%) and macrophages (70%), whereas recombination in LCs (43%), dendritic cells (16%), and mast cells (26%) was less effective. T and B cells were largely unaffected. Effective disruption of the GR gene in neutrophils was further demonstrated by the complete conversion of the GR^{flox} allele into the GR^{null} allele in FACS-sorted EGFP+GR-1+ Granulocytes showing a 92% and 70% efficiency in neutrophils and macrophages.

GR^{Alb-Cre} mice: Southern blot indicated highly efficient recombination efficiency only in the liver, and immunohistochemical staining showed an almost complete loss of GR in hepatocytes (Kellendonk, 2000, genesis 26:151-153). Mueller et al. also showed efficient GR knockout by Western blotting in total liver extracts (Mueller et al., 2011, Hepatology 54(4):1398-409).

Overall, we think that the excellent recombination and knockout efficiency has been well documented, and we have added these relevant references in the text on lines 501-502. Therefore, although it is impossible to fully exclude that minor residual GR expression might have played a role, this seems highly unlikely.

-Fig 5. It is not clear if the ChEA3 analysis was performed with the model's use or if it was performed using databases. Can the authors clarify this? Did they perform the ChIPseq?

We appreciate the referee's comment. To clarify, we did not perform ChIP-seq analysis in this study. Instead, we used CHEA3, an online tool that provides transcription factor enrichment analysis (TFEA) by leveraging publicly available ChIP-seq data from various sources. This tool allowed us to explore the enrichment of transcription factors, including STAT3, in our dataset based on known TF binding to gene promoters as indicated by these existing ChIP-seq databases.

Fig 5F: is the correlation calculated with the pSTAT/pSTAT3 data from the WB on E? WB is a semiquantitative analysis. The authors should use a quantitative method to measure the pSTAT/pSTAT3 and make the correlation.

We appreciate the referee's insightful comment. The correlation shown in Figure 5F was indeed calculated based on the semiquantitative pSTAT3/STAT3 data from the Western blot presented in Figure 5E. While we fully acknowledge that Western blotting is a semiquantitative method, it remains the golden standard method to assess STAT activation. To enhance clarity and transparency, we have now explicitly stated in the relevant figure legends that the band intensities were determined by semiquantitative densitometric analysis.

Fig 6: The absence of phenotype in conditional mice don't match the results in figure 6 where inflammatory cytokines are upregulated. Experiments with WT or KO mice receiving KO or WT BM cells would reinforce the author's conclusions.

Respectfully, we do not fully agree that findings don't match. The GR is ubiquitously expressed in almost all cells of the body, including leukocytes, endothelial cells, hepatocytes and many other cells. The absence of phenotype in individual conditional knockouts suggests that GR deficiency in one of the three cell types tested is insufficient to recapitulate the systemic metabolic and inflammatory disturbances observed in global GRiKO mice. This may indicate that either the GR needs to be knocked out in another cell type (but it is hard to test all possible cell types in this way), or more likely, that the GR affects cytokine expression and/or glucose metabolism in many cell types, leading consequently to hypoglycemia.

We acknowledge that the suggested bone marrow chimera experiments could help distinguish between hematopoietic cells vs non-hematopoietic contributions. However, we presently do not have the expertise and necessary ethical approvals to perform such technically demanding experiments within the limited timeframe of the revision. Furthermore, it is also possible that GR knockout is required in hepatocytes and leukocytes to obtain hypoglycemia, which would then still not be detectable in bone marrow chimera experiments.

In general, anti-inflammatory and other activities of GCs have been proven in a variety of cells beyond the T cells, myeloid cells and hepatocytes investigated in our study. This includes both hematopoietic and non-hematopoietic cells, e.g. endothelial cells (Zielinska et al., Front Immunol 2016), epithelial cells (Muzzi et al., Cell Mol Gastroenterol Hepatol 2020), muscle cells (Braun and Marks, Front Physiol 2015), adipocytes (Lee et al., Biochim Biophys Acta 2014), fibroblasts (Nakamura et al., Scientific Rep. 2020), erythroblasts (Hansen and Iskander, Front Hematol 2025). Similarly, IL-6 (a main cytokine to activate STAT3) can also be produced by a variety of cells, including both hematopoietic (macrophages, lymphocytes) and non-hematopoietic cells (fibroblasts, muscle cells, endothelial cells). This further supports our opinion that the GR in multiple cell types may be responsible for the observed phenotype in our study.

Fig 7: Although the use of Ruxolitinib is innovative and adds a translational aspect to the study, I have several comments in this regard:

- Survival of Ruxolitinib-treated mice compared to GRi ko that don't die
- The glucose uptake experiments, as in Fig 2, should be done
- Immune cells and cytokine expression on the treated mice?
- What is the impact of the Ruxolitinib treatment on other murine malaria models? Pyoelii NL or Pberghei NK65?
- In the T-cell response experiments, the Effector phenotype is assessed, but are these antigen-responding cells?

We would like to thank the referee for these thoughtful comments.

Upon request of the referee, we verified whether there is a significant difference between the Ruxolitinib-treated GRiKO mice and the surviving mice. However, we could not find a significant

difference. We have included this analysis in the graph accordingly (fig. 8B).

Unfortunately, the FDG-PET/MRI facility is not available anymore in our university. As an alternative, we performed a similar experiment by injecting mice with ¹⁸F-FDG and subsequently measuring organ-specific FDG uptake using a gamma counter on dissected tissues. The data revealed that Ruxolitinib treatment does not significantly affect FDG uptake in the liver or spleen (Appendix fig. S11A, S11I), despite its pronounced protective effect against hypoglycemia. Since FDG uptake primarily reflects the early steps of glucose metabolism (i.e., glucose transport and phosphorylation in upper glycolysis), this finding suggests that ruxolitinib exerts its protective effect downstream or via parallel metabolic pathways not captured by FDG-based assays. It is interesting that splenic *Hk1* expression appeared inhibited by ruxolitinib, this may have played a role by inhibiting excessive glycolysis by the spleen (fig. 8G). Alternatively, posttranslational regulatory mechanisms may have been involved. We have further added a few lines about this in the discussion (440-447).

Importantly, we performed a multiplex assay on the plasma samples of WT and GRiKO mice, treated or not with ruxolitinib (fig. 9). The data indicate that ruxolitinib successfully suppresses the excessive cytokine production, particularly of IL-6 and IL-10, in infected GRiKO mice. These findings confirm that ruxolitinib not only alleviates hypoglycemia, but also mitigates systemic hyperinflammation, both of which likely contribute to the improved survival observed in treated GRiKO mice.

Upon request of the referee, we also performed two independent experiments to evaluate the effect of ruxolitinib in an additional mouse model of severe malaria: infection of WT C57BL/6 mice with *PbNK65* (Edinburgh strain) (fig. 10). We selected this model based on prior work from our group (Vandermosten 2023, Front Immunol), which demonstrated that these mice develop GC resistance, thus mimicking the clinical situation observed in severe human malaria. In this study, we first confirmed that STAT3 activation is also present in this model, supporting its relevance for testing JAK/STAT-targeted therapies. Ruxolitinib treatment led to a pronounced reduction in clinical scores, as well as a marked improvement in hypoglycemia and hypothermia. These results indicate that the protective effects of ruxolitinib are not limited to the GRiKO model, but extend to a distinct, GC-resistant model of severe malaria. Together, these findings further support the therapeutic potential and broader applicability of targeting the JAK/STAT3 axis in severe malaria.

Referee #3:

Prenen & Vandermosten et al. gathered an impressive amount of work investigation the role of glucocorticoid receptor in disease tolerance during *Plasmodium chabaudi* infection. Their data show that in the absence of GR induce by tamoxifen in GRiKO infected mice more often develop lethal hypoglycemia, and that their liver and spleen retain glucose. They tested single KO lacking GR in hepatocyte, T-cells, or myeloid cells specifically and observed no increased lethality, but combinations of multiple KO were not tested. The authors then investigated liver and spleen gene expression in on infected and control mice with and without tamoxifen treatment to abolish GR, and

detected the DEGs were associated with JAK/STAT activation and show an increased ratio of phosphorylated STAT3 over total, supporting increased STAT3 activation in GR deficient mice. Then they used Ruxolitinib to inhibit stat3 in infected mice with and without GR and observe a recovery of the lethal impact of Pcc in mice. I believe their efforts are worthy of publication and I have only a few suggestions that I think may help the reader follow and improve possible discussion of future work.

Comments and questions:

Ln 56. what tissues were used to detect the elevated GCs in malaria mice and humans? Be clear

In our study, we did not measure glucocorticoid (GC) levels. However, elevated GC levels in malaria-infected mice and humans have been detected in other studies, specifically in plasma, where GCs circulate. References have been added at line 58-59 in the introduction: (Vandermosten et al, 2018; Abdrabou et al, 2021; Vandermosten et al, 2023). This highlights the systemic nature of GC elevation during malaria.

Ln. 83 I'd mention briefly in the result section how does tamoxifen-inducible glucocorticoid receptor knockout mice work, and I suggest adding Ref in the result and not only in the methods GRiKO mice is maybe not the best name

We have now included a brief explanation in the results section (line 84-89) to describe how tamoxifen-inducible glucocorticoid receptor knockout (GRiKO) mice function, and have added the relevant references in the results sections as well.

Fig. S3c parasitemia could be compared at day of death vs that day of surviving mice, instead of d10.

We chose to present parasitemia data (and in the revised manuscript now also systemic and liver-specific parasite load as well) on day 10, as this time point corresponds to the day when most of the mice developed hypoglycemia. Comparing parasitemia at this critical timepoint provides a consistent and biologically relevant point of analysis, as it reflects the peak of disease severity across the cohort. In theory, we can take the parasitemia at day of death (d10, d11, d12,...), but it is then hard to compare with surviving mice, as it would be an arbitrary decision which mice to take at which time point for the surviving mice.

Is Fig 2b & C done from calculations images in Fig2 A or from the organs removed? Should be clear from reading the result section too.

We appreciate the referee's comment and agree that clarification is needed. The images presented in Figures 2A are representative 18FDG PET/MRI scans. The data shown in Figures 2B and 2C were calculated based on uptake measurements for the regions of interest (liver and spleen) directly from these images. Hence, organs were not removed, but the uptake was determined directly in the living mice. This information has now been clarified in the results section (line 156-157) for improved transparency.

Fig. S5 was done at d10 in sacrificed mice, but seems unclear what their survival outcome would be, was the clinical score recorded? Representative images are shown but how many mice were used, and were there 2 different outcomes 50/50 as would be expected from the clinical score and lethality description earlier.

For the H&E staining, mice were dissected on day 10 post-infection, which precluded the assessment of survival outcomes. At this time point, some GRiKO mice had already developed

hypoglycemia, while others had not. In total, the staining was performed on 20 mice: 3 CON WT, 3 CON GRiKO, 6 *PcAS* WT, and 8 *PcAS* GRiKO (including both normoglycemic and hypoglycemic individuals). We have now added the corresponding glycemia levels for the mice of which representative images are shown in Appendix S3. We included representative images a hypoglycemic infected GRiKO mouse. Additionally, we quantified the histopathology using a scoring system adapted from Knackstedt et al., (2019, *Sci Immunol*). Scoring was performed in a blinded manner by an experienced investigator. Importantly, no differences could be found based on the hypoglycemia state of the mice. All updates are provided in Appendix S3 and the scoring system can be found in Appendix table S3.

Ln 175 & 180 I'd mention that the only single KO and no multiple combinations were not tested

We clarified in the manuscript (line 189) that only single knockouts were tested, and combinations of multiple knockouts were not evaluated. Please note that logistically it would be rather difficult to generate multiple knockouts, as this would need triple genetic modifications (GRflox+2 different Cre-expressing constructs), which would be fairly hard to breed.

Ln 253 I'd mention that ruxolitinib has been shown to inhibit stat3 also, and not only that it is a selective JAK1/2 inhibitor, and I'd show the clinical score on the main Fig7 and not on the Fig. S I'd mention the symptoms that were present when the treatment started, and would discuss how late could the treatment be applied, thinking of future clinical application for children with severe malaria.

We agree and clarified in the manuscript that ruxolitinib acts as a selective JAK1/2 inhibitor and thereby consequently also inhibits STAT3 activation. This strengthens the rationale for its use in targeting the STAT3 pathway in our study. We have emphasized the link between JAK1/2 and STAT3 in the result section (line 277-278 and 285-286).

We also moved the clinical score data appendix to Figure 8. As you can see in the clinical score graph, the symptoms at the start of the treatment are very limited, while parasitemia levels are at circa 5%. The current treatment regimen in this manuscript was designed to mimic an early therapeutic intervention. In the revised manuscript, we now also included data on the efficacy of early ruxolitinib therapy regimen in the *PbNK65-E* model (fig. 9). Nevertheless, we acknowledge the importance of discussing how late ruxolitinib treatment could be applied in a clinical setting. Therefore, we will include a discussion on the potential for later treatment application, considering the acute progression of severe malaria in children. We will also highlight the need for further studies to determine the therapeutic window and its applicability in clinical settings (line 448-456).

3rd Jun 2025

Dear Prof. Van den Steen,

Thank you for submitting your revised manuscript to EMBO Molecular Medicine. We have now received the enclosed report from the three referees who re-assessed your work. I'm pleased to inform you that all referees are now supportive, and we are able to accept your manuscript pending the following editorial-level amendments:

1. Please remove the "Authors' contribution" section from the manuscript file.
2. Delete the running title from the manuscript text.
3. Appendix: please revise the nomenclature to follow the correct format-e.g., "Appendix Figure S1," "Appendix Table S1," etc.
4. Ensure that the funding information is consistent between the submission system and the manuscript file. Specifically, the PhD fellowship from F.W.O.-Vlaanderen is mentioned in the manuscript but missing from the submission system-please update accordingly.
5. Figure callouts:
 - All individual panels for Figures 3 and 6 should be explicitly called out in the text.
 - Please add a callout for Figure 5A.
6. Data availability: please remove the reviewer access code and ensure that the datasets will be made publicly available upon acceptance of the manuscript.
7. The current synopsis image is too large. Please resize to meet the required dimension: PNG format 550 px wide x 300-600 px high.
8. Please remove the synopsis text from the manuscript file and upload it separately as a Word (.doc) file.
9. Please address the following issues related to figure legends:
 - Please note that legends for Figure 9K and 9L are missing-these should be added.
 - Please note that the exact p values are not provided in the legends of figures 1B, C, D, F, G, H, J, K, L, M; 2B, C; 3N, 4C, D, E, F, H, I, J; 5E-J; 6A, C, D; 8B, E, F, G, H, I; 9A, B, C, D, I, J, K, L; 10B,D, F, G, H
 - Please indicate the statistical test used for data analysis in the legend of figure 7C

I look forward to reading a new revised version of your manuscript as soon as possible.

Kind regards
Jingyi

Jingyi Hou
Senior Editor
EMBO Molecular Medicine

*** Instructions to submit your revised manuscript ***

**** Reviewer's comments ****

Referee #1 (Remarks for Author):

The authors have addressed all my comments.

Referee #2 (Remarks for Author):

The authors have not only responded to my questions and concerns but also improved the manuscript.

I therefore recommend publication and congratulate the authors on thoroughly revising their manuscript. I look forward to the next translational studies on this topic.

All editorial and formatting issues were resolved by the authors.

12th Jun 2025

Dear Prof. Van den Steen,

Please find enclosed the final reports on your manuscript. We are pleased to inform you that your manuscript is accepted for publication and is now being sent to our publisher to be included in the next available issue of EMBO Molecular Medicine.

Sincerely,
Jingyi

Jingyi Hou
Senior Editor
EMBO Molecular Medicine
